**STEM CELLS AND REGENERATION**

# The emergence of multiple testicular cell lineages in human stem cell-derived testis-like organoids

Svenja Pachernegg[1,2,3,4,*], Gorjana Robevska[1], Lucas G. A. Ferreira[1], Natalie Charitakis[3,4], Jinchao Gu[5], Jan Terhag[6], Eliza Martin[1], Denis Bienroth[3,4], Jocelyn van den Bergen[1], Sean B. Wilson[4,7], Fernando J. Rossello[3,4,8], Ben Rollo[5], Melissa H. Little[4,7], Mirana Ramialison[2,3,4,9], Andrew H. Sinclair[1,2] and Katie L. Ayers[1,2,*]

## ABSTRACT

Reproductive development is a complex process orchestrated by precise gene expression and cellular interactions. Disruption to this process can result in differences of sex development (DSDs) which occur in approximately 1-2% of live births. We have previously developed a protocol to differentiate human induced pluripotent stem cells (hiPSCs) into testis-like organoids. In this study, we performed bulk and single-cell RNA-sequencing on these organoids to investigate their transcriptional landscape. Transcriptomic analysis revealed six distinct cell clusters expressing markers associated with bipotential, early Sertoli and testicular interstitial cells. These findings provide the first comprehensive transcriptional profile of hiPSC-derived testis-like organoids. Additionally, to address the limited emergence of mature cell types, we generated an inducible *NR5A1*/SF1 hiPSC line, which successfully triggered the upregulation of Leydig cell markers and additional Sertoli markers upon overexpression. Our findings show that our testis-like organoids are a valuable model system for studying DSDs *in vitro*.

KEY WORDS: Gonadal development, Testis development, DSD, Differences of sex development, Inducible SF1 cell line, Transcriptomics

## INTRODUCTION

Reproductive development, including the development and differentiation of the testes and ovaries, is orchestrated by precise temporal sequences of gene expression and cellular interactions (Lundgaard Riis and Jørgensen, 2022). Disruptions to these processes

[1]Reproductive Development, Murdoch Children's Research Institute, Parkville, VIC 3052, Australia. [2]Department of Paediatrics, University of Melbourne, Parkville, VIC 3010, Australia. [3]Transcriptomics and Bioinformatics, Murdoch Children's Research Institute, Parkville, VIC 3052, Australia. [4]Novo Nordisk Foundation Centre for Stem Cell Medicine, Murdoch Children's Research Institute, Parkville, VIC 3052, Australia. [5]Department of Neuroscience, School of Translational Medicine, Monash University, Melbourne, VIC 3004, Australia. [6]Research Support Operations, Murdoch Children's Research Institute, Parkville, VIC 3052, Australia. [7]Novo Nordisk Foundation Centre for Stem Cell Medicine, Faculty of Health and Medical Sciences, University of Copenhagen, 2200 Copenhagen, Denmark. [8]Department of Clinical Pathology, University of Melbourne, Parkville, VIC 3010, Australia. [9]Australian Regenerative Medicine Institute, Monash University, Clayton, VIC 3800, Australia.

*Authors for correspondence (Svenja.Pachernegg@mcri.edu.au; Katie.Ayers@mcri.edu.au)

S.P., 0009-0003-9116-6064; K.L.A., 0000-0002-6840-3186

can cause Differences of sex development (DSDs) which affect up to 2% of babies born (Cools et al., 2018; Delot et al., 2017). The genetic aetiology of more than 50% of DSDs is unclear, in part due to a lack of understanding of human reproductive development and a dearth of human specific *in vitro* models in which to study candidate causes (Cunha et al., 2019; Monaco et al., 2015). Immortalised human cell lines from testicular (NTERA-2 cl.D1 cells; Andrews, 1984) or ovarian (COV434 cells; van den Berg-Bakker et al., 1993) adult malignant tumours have been established, but their ability to accurately recapitulate human fetal gonadal development is limited. Stem cell technologies have opened new avenues for studying human development and its disruption for a variety of tissues [e.g. kidney (Takasato et al., 2015), brain (Lancaster et al., 2013), heart (Lewis-Israeli et al., 2021) and gut (Spence et al., 2011)] and the generation of gonadal cells from human pluripotent cells holds immense promise in understanding the cellular processes governing fetal reproductive development, disease modelling for DSDs, and applications in reproductive medicine.

In humans, the bipotential gonads arise along the urogenital ridge at 5-6 weeks gestation and subsequently differentiate into ovaries or testes (O'Shaughnessy and Fowler, 2014). Briefly, in the embryonic testis, primordial germ cells (PGCs) and supporting Sertoli cells (SCs) reside within the seminiferous tubules, which are enclosed by basal lamina secreted by SCs and interstitial peritubular myoid (PTM) cells. The steroidogenic Leydig cells (LCs) that produce testosterone reside in the interstitial space between tubules. Differentiation of the bipotential gonad into testis is triggered by the expression of the *SRY* (sex-determining region Y) gene (Sinclair et al., 1990). *SRY* together with *NR5A1*/SF1 (nuclear receptor 5A1, encodes steroidogenic factor 1) activates *SOX9* (SRY-box transcription factor 9), which triggers the differentiation of SCs (Hanley et al., 1999, 2000; Luo et al., 1994). LC and PTM cell differentiation is initiated via *NR5A1*/SF1 and *DHH* (desert hedgehog signalling molecule) (Chen et al., 2017; Schimmer and White, 2010).

Over the past 15 years, the field of *in vitro* human testis differentiation has evolved significantly, with several protocols emerging to direct stem cell fate toward testicular somatic cell fates. Directed differentiation approaches using small molecules, growth factors and hormones have been established to generate bipotential cell-like (Sepponen et al., 2017), SC-like (Bucay et al., 2009; Gonen et al., 2023; Rodríguez Gutiérrez et al., 2018; Kjartansdóttir et al., 2015), LC-like (Chen et al., 2019; Kjartansdóttir et al., 2015; Shin et al., 2021) and PTM cell-like (Robinson et al., 2023) cells from human embryonic stem cells (hESCs) and human induced pluripotent stem cells (hiPSCs). More recently, forced expression of *NR5A1*/SF1 through either overexpression (Buganim et al., 2012; Li et al., 2019; Parivesh et al., 2024) or inducible cell systems (Ishida et al., 2021) in combination with the supplementation of growth factors and small

*DEVELOPMENT*

molecules have been used to efficiently differentiate hiPSCs into LC-like cells. Another advancement has been the development of protocols for testis-like organoids including from our lab (Knarston et al., 2020) and others (Pryzhkova et al., 2022). Our directed differentiation protocol for hiPSCs yields testis-like organoids with significant upregulation of key markers of the bipotential gonads and SCs (Knarston et al., 2020). Among these, SOX9 and GATA4 (GATA binding protein 4) exhibit expression in tube-like structures in these 3D models (Knarston et al., 2020). Similarly, Pryzhkova et al. (2022) have differentiated hESCs into testis-like organoids expressing gonadal markers along with SOX9 protein expression in tubular structures. Despite these advances, achieving functional gonadal cells and organoids that fully recapitulate the fetal testis remains challenging, and their transcriptional landscape remains largely unknown.

In recent years, bulk and single cell RNA-sequencing (RNA-seq and scRNA-seq) have greatly enhanced our ability to understand transcriptional changes during development, with several groups applying these technologies to human embryonic and fetal gonads (Garcia-Alonso et al., 2022; Guo et al., 2021; Lecluze et al., 2020; Li et al., 2017; Taelman et al., 2024; Wang et al., 2022). However, in-depth transcriptomic analysis and comparison of hiPSC-derived gonadal- or testis-like organoids have not been performed so far. In this study, we performed RNA- and scRNA-seq on testis-like organoids generated from hiPSCs using our previously established protocol (Knarston et al., 2020). Transcriptomic analysis revealed distinct cellular populations, with lineages including bipotential, early Sertoli and testicular interstitial cells. This cellular heterogeneity suggests that our organoids recapitulate key aspects of gonadal development and cellular organisation. This analysis also revealed that a number of mature Leydig cell markers were absent, and we hypothesised that a lack of sustained endogenous *NR5A1*/SF1 expression hindered the differentiation of these cells. We therefore generated a transgenic doxycycline (DOX)-inducible hiPSC line (PCS_AAVS1-SF1) to force the expression of *NR5A1*/SF1 in our testis-like organoids. This increased expression of mature Sertoli and Leydig cell markers underlines the importance of *NR5A1*/SF1 expression during fetal sex differentiation and in *in vitro* models. Our study provides a transcriptional resource from testis-like organoids and offers insights into the molecular mechanisms underlying human fetal gonad development and differentiation.

## RESULTS
### Bulk RNA-seq confirms bipotential gonadal and early Sertoli cell gene expression in testis-like organoids
We previously developed a protocol to differentiate hiPSCs into early testis-like cells and organoids (Fig. 1A) that show significant upregulation of bipotential and SC markers compared to undifferentiated stem cells (Knarston et al., 2020). To validate these findings and gain deeper insights into global transcriptional changes, we performed bulk RNA-seq on undifferentiated stem cells (*n*=3) and day 18 testis-like organoids (*n*=4) [for raw counts, CPM (counts per million) and log2 fold change of gene expression in organoids, see Table S1]. Multidimensional scaling (MDS) showed clear clustering of samples by cell types (Fig. S1A). Differentially expressed gene (DEG) analysis comparing the testis-like organoids to the undifferentiated stem cells (Fig. 1B) revealed downregulation of a number of key markers of stem cell identity in organoids, including genes such as *SOX2* (SRY box transcription factor 2), *POU5F1* [POU class 5 homeobox 1, encodes octamer-binding transcription factor 4 (OCT4)], *ESRG* (embryonic stem cell related) and *NANOG* (nanog homeobox). Conversely, we observed upregulation of various collagens (*COL1A1*, *COL5A1*, *COL6A1*),

*ACTA2* (alpha smooth muscle actin), *CLDN11* (claudin 11), *DCN* (decorin), *GADD45G* (growth arrest and DNA-damage-inducible protein gamma), *GATA4*, *IGF2* (insulin like growth factor 1), *LUM* (lumican), *NR2F2* (nuclear receptor subfamily 2, group F, member 2; encodes for COUP-TFII), *WT1* (Wilms' tumour gene 1) and *ZFPM2* (zinc finger protein, multitype 2 protein; encodes FOG2), genes associated with gonadal development and function (Ferrari et al., 2022; Ferreira et al., 2024; Garcia-Alonso et al., 2022; Griffeth et al., 2014; Liebich et al., 2022; Mamsen et al., 2020; McClive and Sinclair, 2003; Padua et al., 2015; Stammler et al., 2016; Viger et al., 2022; Wankanit et al., 2024). These expression patterns suggest a transition from pluripotency towards a gonadal cell fate. To gain deeper insights into cell type-specific gene signatures, we analysed the expression of marker genes for testicular cell lineages and found distinct expression patterns of genes associated with bipotential, interstitial and SC fates in the testis-like organoids compared to the undifferentiated stem cells (Fig. 1C,D). Markers for ovarian cells were not expressed (*FOXL2*, forkhead box L2; *WNT4*, Wnt family member 4) or were only weakly expressed (*RSPO1*, R-spondin 1), with the latter also being expressed in undifferentiated gonadal cells in both male and female mice (Chassot et al., 2012).

Interestingly, in this bulk RNA analysis expression levels of *NR5A1*, *AMH* (anti Müllerian hormone)*,* and *SOX9* showed no significant changes between the undifferentiated stem cells and the testis-like organoids (Fig. 1B,C,E). Examination of the normalised expression data (CPM) revealed that *SOX9* showed high expression levels in both undifferentiated stem cells and organoids, explaining the lack of significant upregulation despite its expression in testis-like organoids (Fig. 1E). This suggests that this crucial transcription factor maintains expression levels in both undifferentiated stem cells and testis-like organoids, and that high levels of expression of SOX9 may only be observed in a small subset of cells as we have seen previously in immunofluorescent (IF) staining of organoids after day 18 (Knarston et al., 2020).

To confirm the presence of multiple gonadal cell lineages in our testis-like organoids, we assessed the top 500 upregulated DEGs by filtering for the top 500 genes that were significantly [false discovery rate (FDR)<0.05] upregulated (log2 fold change>5) in the organoids compared to the undifferentiated stem cells (Table S1). We then queried in which cell clusters these DEGs were co-expressed in a previously published transcriptional dataset of human fetal gonadal development (Garcia-Alonso et al., 2022) by using an interactive viewer of the dataset (https://www.reproductivecellatlas.org/gonads/human-main-male/ and https://www.reproductivecellatlas.org/gonads/human-main-female/). DEGs from our dataset were predominantly expressed in the mesenchymal and interstitial cell clusters (clusters 8, 6b and 6c) in the published dataset (Garcia-Alonso et al., 2022), suggesting that our organoids recapitulate key aspects of these cell populations (Fig. S2A,B). Furthermore, gene ontology (GO) term enrichment (Ashburner et al., 2000; Gene Ontology Consortium et al., 2023; Thomas et al., 2022) analysis of these DEGs revealed multiple terms, including 'male sex differentiation' (GO:004661). We found that 17 out of 168 genes from the 'male sex differentiation' parent gene list were differentially expressed in our organoids (Fig. S3B) and that these genes are primarily expressed in cluster 8 (mesenchymal cells) in the Garcia-Alonso dataset (Garcia-Alonso et al., 2022) (Fig. S2B). Notable among these genes were *FSHR* (follicle stimulating hormone receptor), *GATA4*, *GATA6* (GATA binding protein 6), *PDGFRA* (platelet-derived growth factor alpha), *TCF21* (transcription factor 21) and *WT1*, all of which play crucial roles in testis development (Bhartiya et al., 2021; Ferrari et al., 2022; Garon et al., 2017; Lotfi et al., 2021; Padua et al., 2015; Viger et al., 2022). Additionally, when

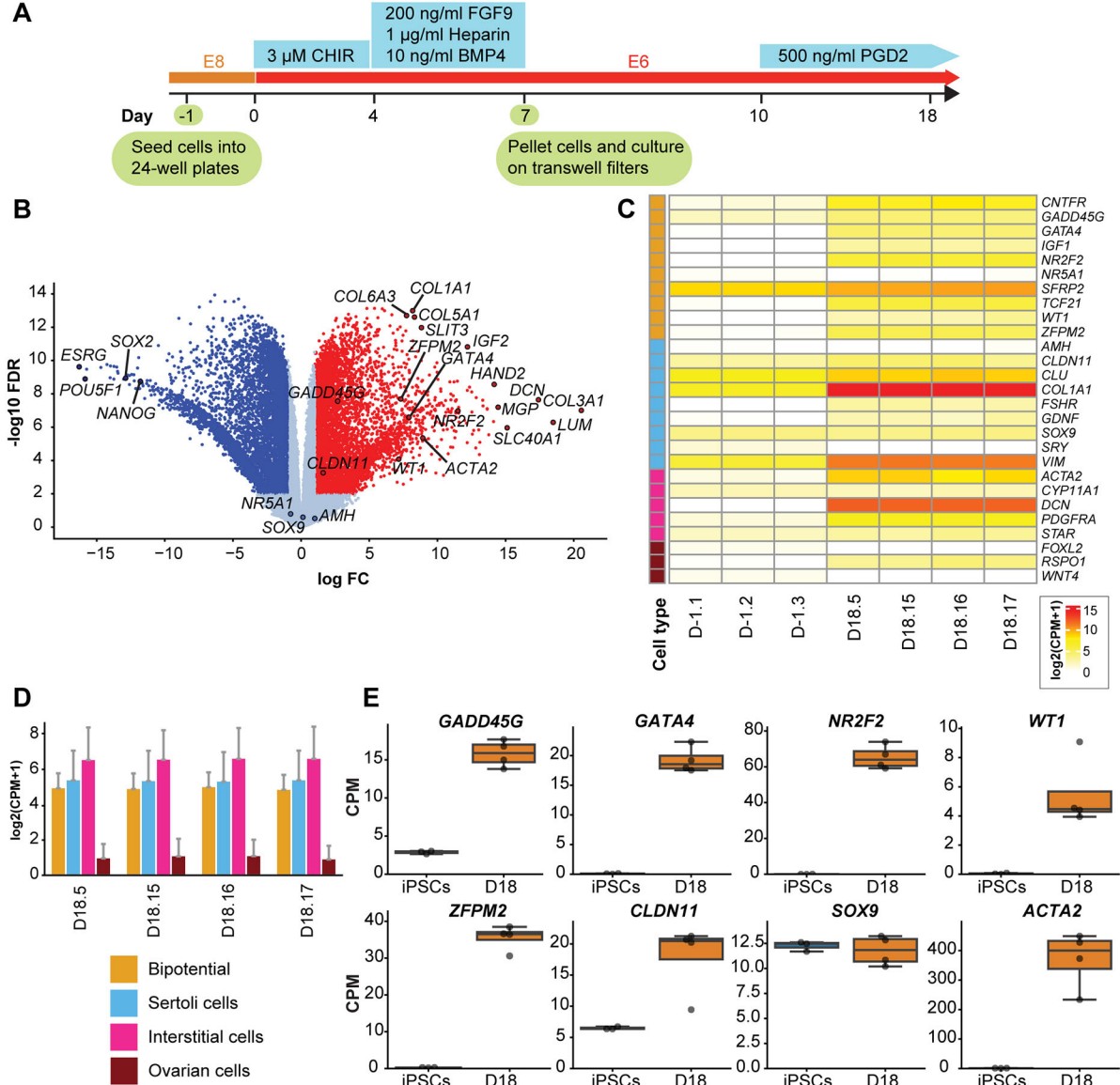

**Fig. 1. RNA-seq confirms bipotential gonadal and early Sertoli cell gene expression in testis-like organoids.** (A) Schematic of the testis-like organoid differentiation protocol (Knarston et al., 2020). (B) Volcano plot of differentially expressed genes (DEGs) in testis-like organoids compared to undifferentiated stem cells. Key genes are annotated. (C) Heatmap showing the expression of testicular marker genes across the undifferentiated stem cell (D-1.1–D-1.3) and testis-like organoid samples (D18.5–D18.17). In the organoid samples, bipotential (orange), SC (blue) and interstitial (pink) markers are upregulated, while ovarian markers (dark red) show absent or minimal expression. (D) Average expression scores of testis and ovarian marker genes, calculated as log2-transformed CPM+1, show the emergence of bipotential, SC and interstitial cell lineages in the testis-like organoids. (E) Normalised expression (CPM) of testis markers in the undifferentiated hiPSCs (blue) and day 18 (D18) organoids (orange). Of note, *SOX9* is also expressed in the undifferentiated stem cells. Box plots show median value (middle bars), first to third interquartile ranges (boxes); whiskers indicate 1.5× the interquartile ranges; dots indicate individual data points; dots outside the whiskers are outliers.

examining an 'immature SC' parent gene list (bioinfo.uth.edu/webcsea), we observed differential expression of 18 out of 19 genes in our organoids, including *COL1A2*, *GATA4*, *GATA6* and *WT1* (Fig. S3B). These genes are expressed in cluster 5 (SCs) in the published testicular dataset (Garcia-Alonso et al., 2022) (Fig. S2B) and to a lesser extent in the granulosa cells in the ovarian dataset (Fig. S2A). This supports the hypothesis that our organoids are progressing along an early gonad and testis developmental trajectory. Notably, the expression of these gonadal markers across both male and female datasets demonstrates the common developmental pathways and bipotential characteristics of early gonadal formation. However, the expression of genes from the 'immature SC' parent gene list is more pronounced in the male dataset in the SC cluster.

## Single cell transcriptomics of testis-like organoids reveals separate clusters expressing sex differentiation and Sertoli cell markers

We postulated that cellular heterogeneity within our organoids may explain why bulk RNA-seq did not show significant upregulation of some key testicular markers despite protein IF analysis showing populations expressing these (e.g. SOX9; Knarston et al., 2020). We also hypothesised that by extending the differentiation protocol for three additional days, we might observe more mature testis cell markers. Therefore, we performed scRNA-seq on two pooled day 21 organoids and sequenced 9577 cells. After data filtering, our final dataset comprised 4928 high-quality cells suitable for downstream analysis (Fig. S1B,C). Six cell clusters were observed

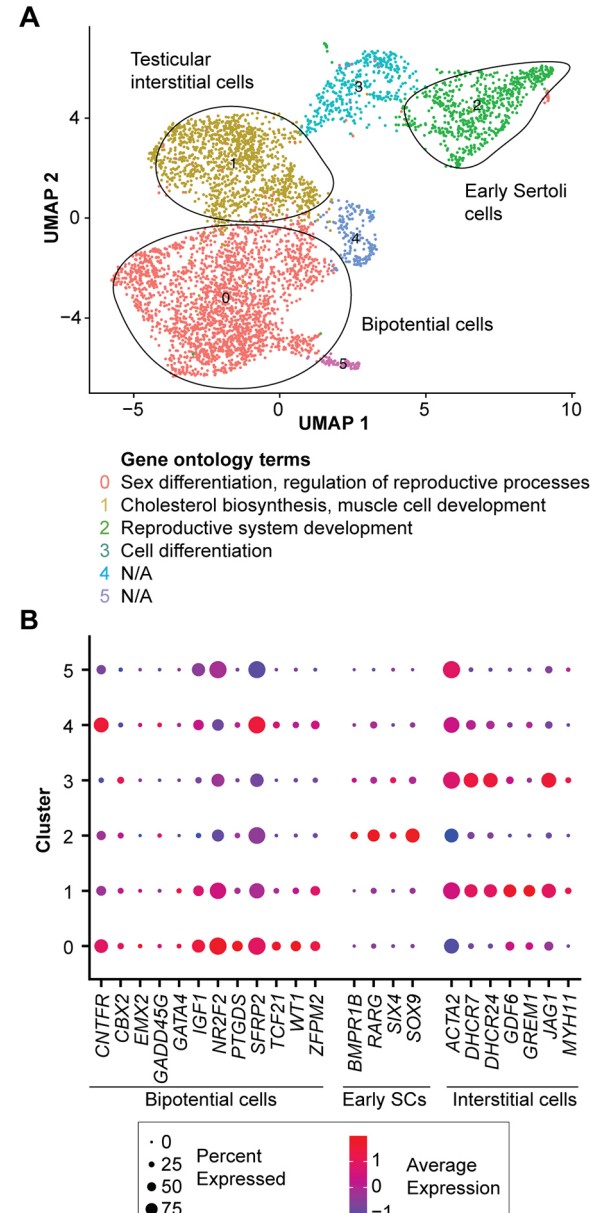

**Fig. 2. Single cell transcriptomics of testis-like organoids reveals separate clusters expressing sex differentiation, Sertoli cell and testicular interstitial cell markers.** (A) scRNA-seq of day 21 testis-like organoids revealed six cell clusters. GO terms associated with the clusters are indicated. (B) Analysis of testis cell markers reveals the presence of bipotential cells, early SCs and interstitial cells (early LCs and PTM cells).

(Fig. 2A; Table S2 for gene lists). Cluster 0 was associated with the GO terms 'sex differentiation' (GO:0007548) and 'regulation of reproductive processes' (GO:2000241). Amongst the DEGs in this cluster were *FGF9* (fibroblast growth factor 9), *GATA4*, *IGF1*, *NR2F2*, *PAX8* (paired box 8), *PTGDS* (prostaglandin D2 synthase), *SFRP2* (secreted frizzled related protein 2), *TCF21*, *WT1* and *ZFPM2*, all of which have been previously associated with fetal gonadal development (van den Bergen et al., 2020; Croft et al., 2023; Ferrari et al., 2022; Ferreira et al., 2024; Gao et al., 2024; Griffeth et al., 2014; Lotfi et al., 2021; Mamsen et al., 2020; Padua et al., 2015; Taelman et al., 2024; Viger et al., 2022; Wankanit et al., 2024; Warr et al., 2009). These genes are co-expressed in clusters 3,

6c and 8 (early supporting, testicular interstitial and mesenchymal cells) in the Garcia-Alonso dataset (Garcia-Alonso et al., 2022) (Fig. S3A). Cluster 1 was associated with the GO term 'cholesterol biosynthetic process' (GO:0006695), with upregulation of genes such as *DHCR7* (7-dehydrocholesterol reductase) and *DHCR24* (24-dehydrocholesterol reductase), key markers of steroidogenesis in early LCs (Anbalagan et al., 2004). Cluster 1 was also associated with 'muscle cell development' (GO:0051147), expressing genes such as *BMP2* (bone morphogenetic protein 2), *BMP6* (bone morphogenetic protein 6) and *ACTA2*. Genes from cluster 1 are co-expressed in clusters 7, 8 and 12 (perivascular, mesenchymal cells and fetal LCs) in the Garcia-Alonso dataset (Garcia-Alonso et al., 2022) (Fig. S3A). Cluster 1 also expressed *CALD1* (caldesmon 1), *DCN*, *DES* (desmin), *IGF1*, *MYH11* (myosin heavy chain 11) and *PDGFRA*, known markers of interstitial early LC and PTM cells (Adam et al., 2012; Garon et al., 2017; Griffeth et al., 2014; Liebich et al., 2022; Mamsen et al., 2020). Cluster 2, associated with the GO term 'reproductive system development' (GO:0061458), showed robust expression of *BMPR1B* (BMP receptor 1B), *RARG* (retinoic acid receptor gamma), *SIX4* (sine oculis homeobox homolog 4) and *SOX9*, as well as *COL2A1*, which are expressed in differentiating SCs (Ciller et al., 2016; Fujimoto et al., 2013; McClive and Sinclair, 2003; Rahmoun et al., 2017). Indeed, these genes are predominantly co-expressed in cluster 5 (SCs) (Garcia-Alonso et al., 2022) (Fig. S3A). Of note, cluster 3 was associated with the broad GO term 'cell differentiation' (GO:0030154), and clusters 4 and 5 expressed too few marker genes to yield statistically significant results in the GO analysis.

Overall, this comparative analysis approach revealed considerable alignment between our testis organoid cellular clusters and mesenchymal, early supporting, SC and testicular interstitial cells in human developing gonads. We then wanted to verify the emergence of different fetal testicular cell types in our testis-like organoids. Specific investigation of well-known marker genes for testicular cell types in our dataset revealed abundant expression of markers associated with bipotential gonad cells, early SCs and interstitial cells (PTM cells and early LCs) (Figs 2B and 3), including *CBX2* (chromobox protein homolog 2), *GADD45G*, *GATA4*, *EMX2* (empty spiracles homeobox 2), *LHX9* (LIM homeobox 9), *NR0B1* (nuclear receptor subfamily 0 group B member 1; encodes for DAX1), *SOX9*, *WT1* and *ZFPM2*, suggesting that our organoids recapitulate key cellular populations of the developing human testis. However, we did not observe expression of *SRY* or of mature SC markers, i.e., *AMH* or *DHH*, in our organoids (Fig. 3). When we looked at the expression of *GATA4* and *WT1* in the subset of *SOX9*-expressing cells (SOX9 expression threshold >0.5 log2-normalised counts) (Fig. S4A), we saw that both were indeed co-expressed within this cell population. Similarly, when we subset for *GATA4* (Fig. S4B) or *WT1* (Fig. S4C) expression (expression threshold for both *GATA4* and *WT1* >0.5 log2-normalised counts), we saw co-expression of SOX9 in some of these cells. Notably, the overlap between *SOX9*, *GATA4* and *WT1* expression was not complete, which was consistent with *in vivo* studies showing variable co-expression of *GATA4* and *SOX9* during early gonadal development at 6-7 weeks post-fertilisation (Guo et al., 2021).

Furthermore, we investigated the expression of a developmental gene panel (Fig. S5). Our differentiation protocol drives cells through an intermediate mesodermal progenitor stage, and both the coelomic epithelium (CE) and the mesonephros are mesodermal structures. However, we found that our organoids did not express markers of mature mesonephros or podocytes (*LHX1*/LIM1, *NPHS1*, *NPHS2*, *PAX2*), nor did they express *GATA2*, which is expressed in mesonephric/epididymal stromal cells but not in

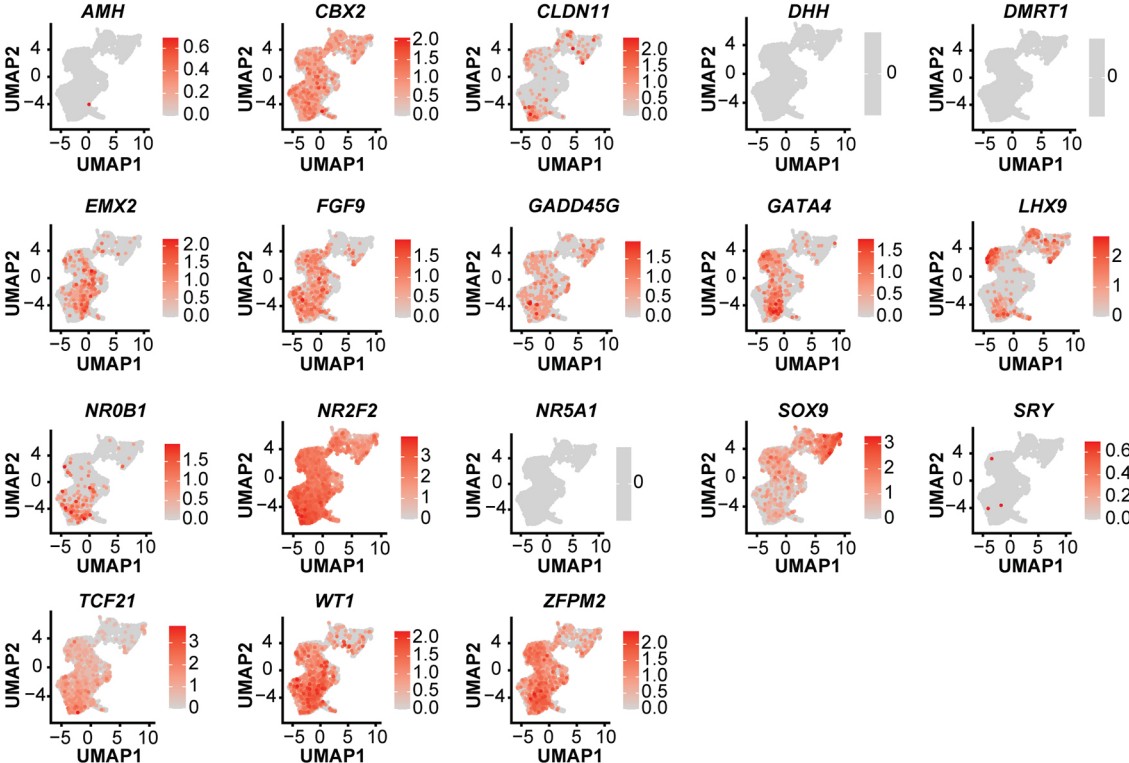

**Fig. 3. Log-normalised expression of gonadal developmental markers in testis organoids.** Feature plots showing expression of key marker genes including *CBX2*, *EMX2*, *GADD45G*, *GATA4*, *LHX9*, *NR0B1*, *NR2F2*, *TCF21*, *WT1* and *ZFPM2* across organoids. While *SOX9* is expressed, the testis markers *AMH*, *DHH*, *DMRT1*, *NR5A1* and *SRY* show minimal or absent expression.

gonadal stromal cells (Taelman et al., 2024). Additionally, classical CE markers *CALB2*, *LRRN4* and *UPK3B* showed minimal expression, with only slight *KRT7* expression detected. However, we did observe expression of *WT1*, *PAX8*, *CXCL14* and *CXCR4*, suggesting the presence of a supporting cell population that may represent an intermediate or gonadal-specific mesodermal state rather than extra-gonadal populations (Garcia-Alonso et al., 2022). Notably, the expression of *PAX8* across all clusters, together with co-expression of WT1, could also be an indication of the intermediate mesodermal origin of our induced cell populations, similar to what has been observed during kidney development (Bouchard et al., 2002).

A goal of developing a testis-like organoid is to apply it to disease modelling. DSDs can be caused by a number of genes, which are collated in the PanelApp DSD gene list (https://panelapp-aus.org/). Our testis-like organoids showed robust differential expression of 18 DSD genes expressed in the bulk RNA and 30 DSD genes expressed in the scRNA-seq data (Fig. S3B,C), indicating their potential as an *in vitro* model for DSD. The DSD panel include genes responsible for conditions where gonadal development or function is perturbed [e.g. *AMHR2* (AMH receptor type 2), *CYP21A2* (cytochrome P450 family 21 subfamily A member 2), *FSHR*, *MYRF* (myelin regulatory factor) and *WT1*; Bramble et al., 2016; Brunello and Rey, 2022; Ferrari et al., 2022; Hamanaka et al., 2019; Kara et al., 2024] and a number of genes that cause DSD conditions where the direct role of the gene in the developing gonads is as yet unclear [e.g. *CDKN1C* (cyclin-dependent kinase inhibitor 1C), *IGF2* (insulin like growth factor 2), *NR3C1* (nuclear receptor subfamily 3, group C, member 1) and *SAMD9* (sterile alpha motif domain containing 9); Audí et al., 2018; Leitao Braga et al., 2022; Narumi et al., 2016; Xie et al., 2022]. In these cases, our testis-like organoid protocol could provide a novel

model in which to better understand the molecular mechanisms underlying these conditions and the variants that cause them. However, our testis-like organoids do not show consistent or sustained upregulation of some mature SC markers, such as *AMH*, *DHH*, or of a number of steroidogenic genes such as *CYP11A1* (cytochrome P450 family 11 subfamily A member 1), *CYP17A1* (cytochrome P450 family 17 subfamily A member 1) or *STAR* (steroidogenic acute regulatory protein), which are present in mature LCs. We hypothesised that this may be a result of inconsistent or unsustained *NR5A1*/SF1 expression in the organoids.

**Overexpression of *NR5A1*/SF1 triggers the expression of Leydig cell markers in testis-like organoids**

*NR5A1* is a crucial transcription factor for gonadal development (Cools et al., 2012; Hanley et al., 1999) and the lack of consistent or sustained expression of this gene in our organoids may explain why we do not see a number of markers of mature SC or LCs. Indeed, we often observe peak expression of *NR5A1* around day 7 of our gonadal differentiation protocol, with subsequent downregulation in testis-like organoids as observed in qPCRs (Knarston et al., 2020), bulk RNA-seq (Fig. 1C) and scRNA-seq (Fig. 3A). Therefore, we tested additional small molecules and hormones in our differentiation protocol to induce sustained *NR5A1* expression (Chen et al., 2019; Kulcenty et al., 2015; Lasala et al., 2011). However, we did not observe upregulation of *NR5A1* in day 18 organoids following treatment with 8-Br-cAMP (8-bromoadenosine 3′-5′-cyclic monophosphate) or with LH (luteinizing hormone), SAG (smoothened agonist), LiCl (lithium chloride) and 22R-OHC (22R-hydroxycholesterol) either alone or in combination (Fig. S6A). We then generated a DOX-inducible *NR5A1*/SF1 hiPSC line (PCS_AAVS1-SF1) by targeting the AAVS1 safe harbour site with

a transgene cassette using CRISPR/Cas9 technology. This PCS_AAVS1_SF1 hiPSC line maintained expression of the key pluripotency markers OCT3/4 and SOX2 in the undifferentiated state (Fig. 4A). To verify trilineage differentiation potential, we generated embryoid bodies from PCS_AAVS1_SF1 hiPSCs and assessed the expression of lineage-specific markers. We observed robust expression of MAP2 (microtubule-associated protein 2; ectodermal marker), SOX17 (endodermal marker) and SMA (smooth muscle actin; mesoderm marker) (Fig. 4B). Functional validation of the inducible system showed significant upregulation of *NR5A1*/SF1 in mRNA and protein expression after treatment with DOX for 48 h (Fig. 4C,D).

To test whether sustained upregulation of *NR5A1*/SF1 in our testis-like organoids induced the expression of mature SC and LC markers, we differentiated the PCS_AAVS1_SF1 hiPSCs into testis-like organoids following our established protocol (Knarston et al., 2020), while simultaneously inducing *NR5A1*/SF1 expression via DOX treatment from differentiation day 2. We sampled the organoids at day 18 to determine whether *NR5A1*/SF1 induction could enhance Leydig or SC marker expression earlier in the differentiation process compared to our previous protocol. After 18 days of culture, qPCR analysis showed that DOX treatment successfully induced high levels of *NR5A1* RNA expression (Fig. 5). This sustained *NR5A1* overexpression resulted in significant upregulation of bipotential gonad markers and SC-specific genes compared to untreated control testis-like organoids (Fig. 5). Notably, the mature SC markers *AMH*, *DHH* and *INHBB* (inhibin subunit beta b) were upregulated in the DOX-treated testis-like organoids, indicating progression to more mature cells. Moreover, the overexpression of *NR5A1*/SF1 also triggered an increased expression of multiple LC markers. Key steroidogenic enzymes including *CYP11A1*, *CYP17A1*, *HSD3B1* [hydroxy-delta-5 steroid dehydrogenase, 3 beta and steroid delta isomerase 1], *HSD3B2* and *STAR* (Galano et al., 2021; de Mattos et al., 2022; Rebourcet et al., 2019) were significantly upregulated. IF staining confirmed the expression of SF1 protein in the DOX-treated

testis-like organoids (Fig. 6A,B), alongside increased expression of bipotential and SC markers (DHH, GATA4, SOX9) (Fig. 6A,C; Fig. S6B). Importantly, we also observed increased STAR protein expression in the SF1-overexpressing organoids in a separate population to DHH (Fig. 6C). Co-staining of SC markers GATA4, SOX9 and DHH revealed that high SOX9 expression was seen in a subset of GATA4-expressing cells and that, within these, differences in subcellular localisation was observed. Of note, DHH expression appeared to be correlated with cells where SOX9 was predominantly nuclear (Fig. 6D, arrows versus arrowheads).

Together, our results provide a comprehensive transcriptional profile of testis-like organoids generated from hiPSCs. Transcriptomic analysis revealed six distinct cell clusters, with clusters associated with sex differentiation and reproductive system development showing significant similarities to published human fetal testis datasets. Using an inducible *NR5A1*/SF1 hiPSC line, we successfully triggered the expression of mature SC and LC markers, demonstrating that sustained upregulation of SF1 is required to push organoids to a more mature fate. Our findings provide insights into the molecular composition of stem cell-derived testis-like organoids and offer a potential strategy for generating more mature gonadal cell types from human stem cells.

## DISCUSSION

In this study, we performed bulk RNA-seq and scRNA-seq on testis-like organoids derived from hiPSCs following our previously established differentiation protocol (Knarston et al., 2020). Both datasets reveal expression patterns characteristic of fetal testis cells, namely bipotential cells, early SCs and interstitial cells, including PTM cells and early LCs. Indeed, scRNA-seq demonstrated that the largest distinct population of cells (cluster 0) expresses genes associated with gonadal development and sex differentiation, such as *FGF9*, *GATA4*, *PBX1*, *SFRP2*, *WT1* and *ZFPM2*/FOG2, as well as *NR2F2* and *TCF21* (van den Bergen et al., 2020; Croft et al., 2023; Eozenou et al., 2019; Ferrari et al., 2022; Ferreira et al., 2024;

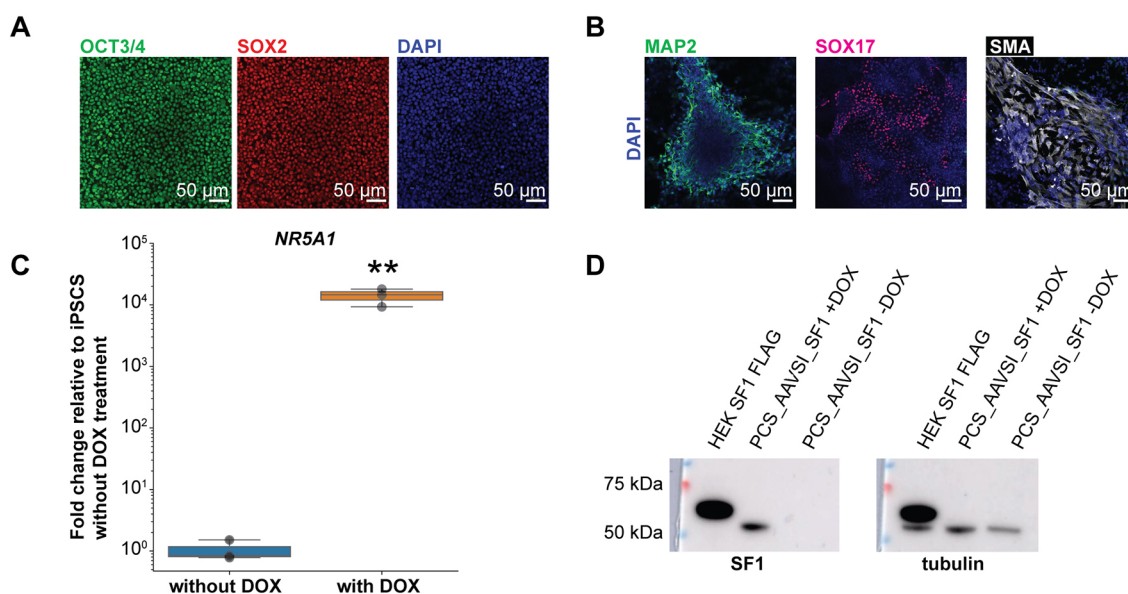

Fig. 4. Cell line validation of the PCS_AAVSI_SF1 hiPSC line. (A) Expression of the pluripotency markers OCT3/4 (green) and SOX2 (red) was confirmed by immunofluorescence staining. DAPI (blue) stains nuclei. (B) The PCS_AAVS1_SF1 iPSCs can differentiate into the three germ layers after embryoid body formation. Immunofluorescence staining shows the expression of the ectodermal marker MAP2 (green), the endodermal marker SOX17 (magenta) and the mesodermal marker SMA (white). DAPI (blue) stains nuclei. (C) After 48 h of treatment with DOX, the PCS_AAVSI_SF1 iPSCs show significant upregulation of *NR5A1* mRNA compared to untreated PCS_AAVSI_SF1 iPSCs. (D) Protein expression of SF1 in DOX-treated PCS_AAVSI_SF1 iPSCs was confirmed via western blot. Scale bars: 50 μm.

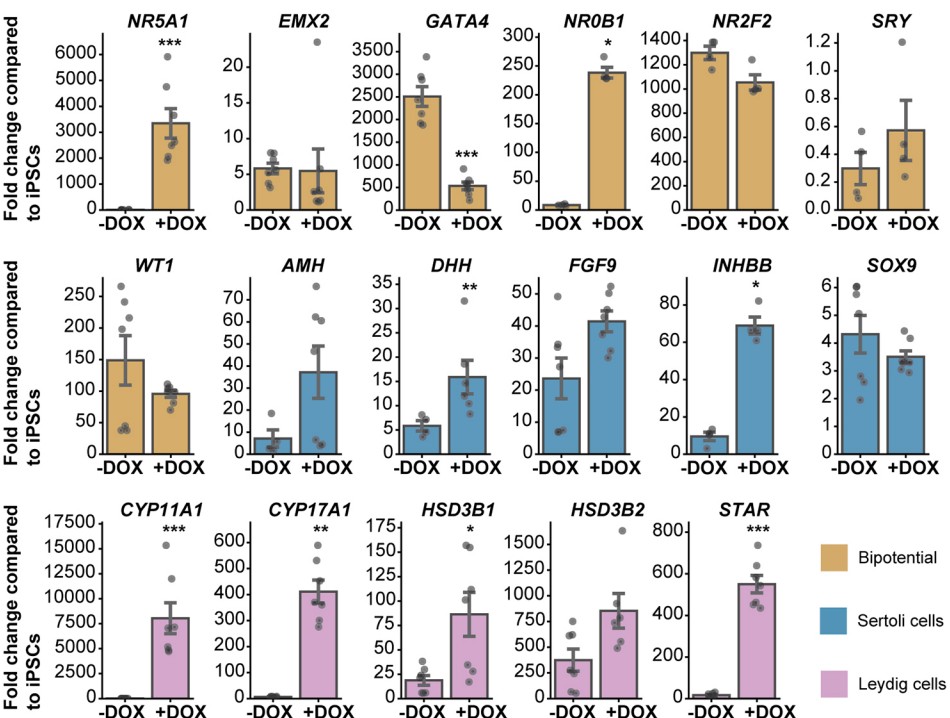

**Fig. 5. Overexpression of *NR5A1*/SF1 triggers the upregulation of Leydig cell and Sertoli cell marker RNA in testis-like organoids.** Relative gene expression of bipotential gonad markers (orange), SC markers (blue) and LC markers (pink) in day 18 testis-like organoids generated with the DOX-inducible PCS_AAVS1_SF1 hiPSC line with or without DOX treatment to induce *NR5A1*/SF1 overexpression. Data represent mean±s.e.m., *n*=2 biological replicates with 3-4 technical replicates. *$P<0.05$; **$P<0.01$, ***$P<0.001$ (Mann-Whitney *U*-test).

Lotfi et al., 2021; Padua et al., 2015; Viger et al., 2022; Warr et al., 2009). Both *TCF21* and *NR2F2/*COUP-TFII are expressed in fetal human testes, where *NR2F2*/COUP-TFII interacts with *NR5A1*/SF1 to regulate *INSL3* (insulin expression in LCs; Di-Luoffo et al., 2022; Lotfi et al., 2021; Lottrup et al., 2014; Wankanit et al., 2024).

During bipotential gonad development, GATA4 and *ZFPM2*/FOG2, together with *WT1* and *NR5A1*/SF1 activate *SRY* and, subsequently, *SOX9* (Tevosian et al., 2002). *SOX9* then maintains its own expression through a positive feedback loop with *FGF9* during testis determination (Jakob and Lovell-Badge, 2011). Of note, our testicular differentiation system operates independently of the Y chromosome and *SRY*. We have previously demonstrated successful *SOX9* expression and testicular differentiation in female hiPSC lines where *SRY* is absent (Knarston et al., 2020). This Y chromosome independence is achieved by bypassing the traditional *SRY*-initiated cascade through direct FGF9 supplementation, which activates *SOX9* and maintains testicular development through auto-regulation mechanisms involving factors such as PGD2 (Moniot et al., 2009), which is added to our differentiation media from day 10 onwards. Indeed, we do not see SRY expression in the organoids at day 18 or 21, consistent with differentiation protocols from other groups who have shown that SRY expression peaks earlier, within the first 84 h of differentiation (Houzelstein et al., 2024). This Y chromosome independence makes our model particularly valuable for studying DSDs where the typical *SRY*-initiated cascade may be disrupted. Interestingly, our current findings show that forced *NR5A1*/SF1 overexpression does result in a small increase in *SRY* expression, consistent with the established role of SF1 in *SRY* regulation, though this upregulation is not required for successful testicular differentiation in our system.

Our organoids also expressed several additional genes that are crucial for early gonadal development, including *CBX2*, *EMX2*, *LHX9* and *NR0B1*. *CBX2* has been identified as essential for normal human male gonadal development and may lie upstream of *SRY* in the sex development cascade (Sproll et al., 2018). During human development, *LHX9* and *EMX2* are expressed in the genital ridge

during the fourth and fifth post-conception weeks, with *EMX2* showing expression in both the XY and XX gonad (Cheng et al., 2022; Ostrer et al., 2007). *NR0B1*/DAX-1 mutations cause adrenal hypoplasia congenita and hypogonadotropic hypogonadism, showing its importance in reproductive development (Achermann and Vilain, 1993). Mutations in these genes (*CBX2*, *EMX2*, *LHX9*, *NR0B1*) have been associated with various DSDs, including gonadal dysgenesis. The expression of this broader panel of gonadal development genes in our organoids suggests that our system recapitulates key aspects of early human gonadal specification.

The presence of markers associated with differentiating SC, particularly *COL2A1* and *SOX9* in cluster 2, further indicates testicular somatic cell differentiation (McClive and Sinclair, 2003; Rahmoun et al., 2017). *COL2A1* is a direct target of *SOX9* (Bell et al., 1997) and is a crucial component of the developing fetal testis cords (McClive and Sinclair, 2003). Cluster 2 also showed high expression of *SIX4*, which has been shown to regulate male gonadal development by regulating SF1 and FOG2 expression in the genital ridges (Fujimoto et al., 2013).

Finally, the association of cluster 1 with genes involved in cholesterol biosynthesis, a hallmark feature of steroidogenic cells, points at the emergence of interstitial cells such as early LCs and PTM cells in our organoids. This is further supported by expression of gene such as *ACTA2*, *CALD1*, *DCN*, *DES*, *IGF1*, *MYH11* and *PDGFRA* (Adam et al., 2012; Garon et al., 2017; Griffeth et al., 2014; Liebich et al., 2022; Mamsen et al., 2020; Taelman et al., 2024). Both *PDGFRA* and *IGF1* are expressed in fetal and adult LCs (Basciani et al., 2010; Griffeth et al., 2014; Mamsen et al., 2020), and *DCN*, an extracellular matrix component, is expressed in interstitial PTM cells, where it interacts with the *PDGF/PDGFR* signalling pathway (Adam et al., 2012). *ACTA2*, *CALD1*, *DES* and *MYH11* have all been shown to be expressed in adult human PTM cells (Liebich et al., 2022).

Cellular identity within our organoids also aligns well with those of human fetal gonads (Garcia-Alonso et al., 2022); indeed, we found that gonadal genes differentially expressed in our dataset are

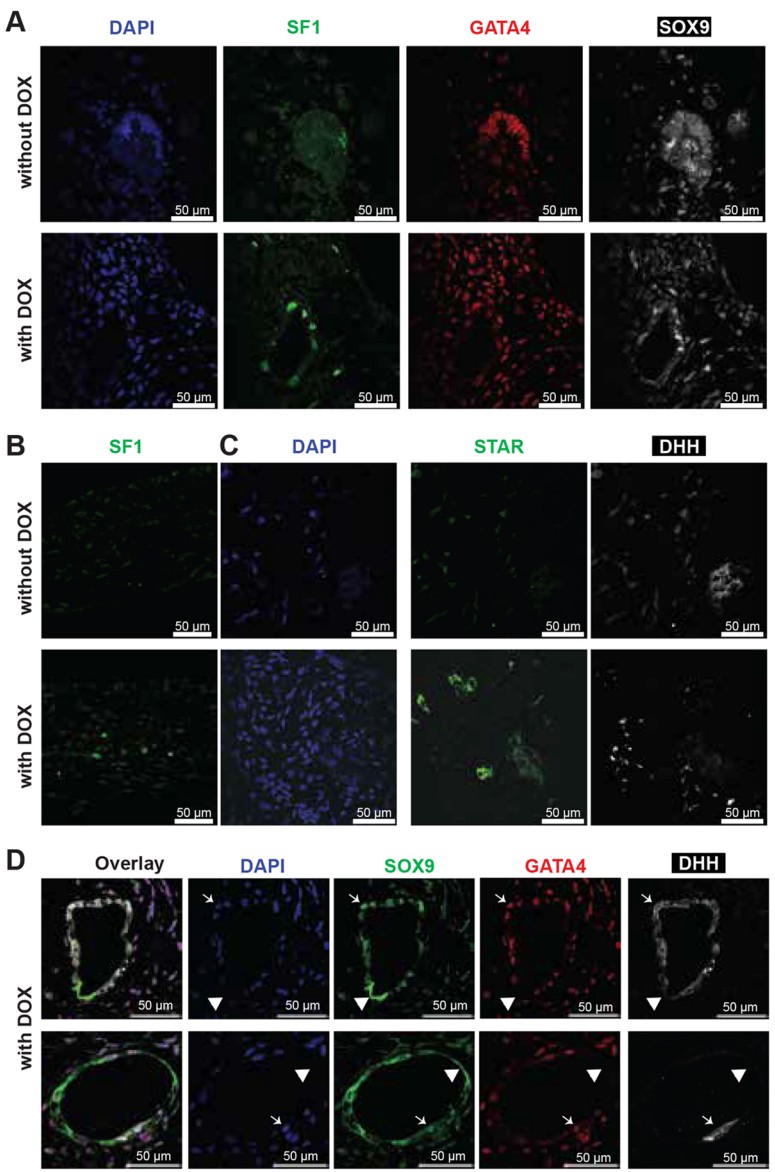

**Fig. 6. SF1-overexpressing organoids express testicular proteins.** (A-D) Immunofluorescence analysis of protein expression in testis-like organoids generated from PCS_AAVS1_SF1 hiPSCs with DOX treatment to induce *NR5A1*/SF1 overexpression and in untreated control organoids. DAPI stains nuclei. (A,B) SF1 protein expression in DOX-treated testis-like organoids in whole (A) and sectioned organoids (B). (A,C) Expression of bipotential gonad (GATA4), SC (SOX9, DHH) and LC (STAR) markers in SF1-overexpressing organoids compared to untreated controls. (D) DOX-induced organoid sections showing co-staining of SOX9, GATA4 and DHH (with DAPI). Cells where SOX9 is predominantly cytoplasmic (arrowhead) show low DHH expression, with arrows indicating cells with predominantly nuclear SOX9 and significant DHH expression. Scale bars: 50 µm.

predominantly expressed in mesenchymal cell and early somatic supporting cell clusters in both female and male datasets, with a proportion of genes expressed in the SC and fetal LC clusters of the human fetal testis. The incomplete overlap between *SOX9* and *GATA4* expression in our scRNA-seq data and immunostainings likely reflects the heterogeneity of SC populations at different maturation states, as *GATA4* is expressed earlier during gonadal development and maintains broader expression, while *SOX9* becomes more restricted to differentiated SCs. Similarly, partial overlap between *GATA4* and *SOX9* expression has been observed in human fetal testis at 6-7 weeks post-fertilisation (Guo et al., 2021). Our immunostainings further revealed that DHH-expressing cells show high levels of both SOX9 and GATA4, and that high DHH expression is restricted to cells with predominantly nuclear SOX9 localisation. This subcellular heterogeneity of SOX9 has been reported previously and may reflect different functional states within the SC lineage (Stewart et al., 2020). Nuclear localisation of SOX9 is required for its transcriptional activity, suggesting that DHH-expressing cells represent a transcriptionally active SC subpopulation, whereas cells with cytoplasmic SOX9 may represent a different maturation state.

Of note, while *PAX8* expression across all cell clusters could potentially indicate the presence of an sPAX8-like supporting cell population (Garcia-Alonso et al., 2022), the widespread expression could also reflect the intermediate mesodermal origin of gonadal precursors. The combination of *PAX8* with gonadal-associated markers (*CXL14*, *CXCR4*) rather than extra-gonadal coelomic epithelium markers (*CALB2*, *LRRN4*, *UPK3B*) suggests that our cells represent a gonadal-biased mesodermal state. Similarly, during kidney development, *PAX8* is widely expressed in the intermediate mesoderm and kidney progenitor cells (Bouchard et al., 2002).

However, despite an increased expression of early testicular cell markers, we did not observe consistent *NR5A1* upregulation in our day 18 or 21 organoids compared to undifferentiated stem cells. *NR5A1*/SF1 plays a crucial role in gonad development and differentiation (Cools et al., 2012; Hanley et al., 1999). To address this, we overexpressed *NR5A1*/SF1 in the testis-like organoids by using a transgenic DOX-inducible hiPSC line. Our engineered PCS_AAVSI_SF1 hiPSC line maintains pluripotency and trilineage differentiation capacity, and showed significant upregulation of *NR5A1*/SF1 after DOX treatment. This drove upregulation of mature SC (*AMH*, *DHH*) and LC (*CYP11A1*, *CYP17A*, *HSD3B1*, *STAR*)

markers, demonstrating the utility of our inducible *NR5A1*/SF1 cell line as a valuable tool for generating more mature testis-like organoids with enhanced steroidogenic potential.

These findings align with previous research demonstrating the potential of exogenous *NR5A1*/SF1 expression to reprogramme mouse and human fibroblasts and iPSCs into SC-like and LC-like cells (Buganim et al., 2012; Li et al., 2019; Liang et al., 2019; Parivesh et al., 2024), either in combination with other transcription factors such as *WT1*, *DMRT1* (doublesex and mab-3 related transcription factor 1), *GATA4*, *GATA6* and *SOX9* (Buganim et al., 2012; Liang et al., 2019; Parivesh et al., 2024), or in combination with small molecules and hormones (Li et al., 2019). Others (Ishida et al., 2021) have also used a DOX-inducible *NR5A1*/SF1 hiPSC line together with 8-Br-cAMP and Forskolin to differentiate LC-like cells which expressed *CYP11A1*, *CYP17A1*, *HSD3B2* (hydroxy-delta-5 steroid dehydrogenase, 3 beta and steroid delta isomerase 2) and *STAR*. Of note, we also saw increased expression of mature SC markers in our testis-like organoids. Interestingly, we observed downregulation of *NR2F2*, mimicking observed downregulation in interstitial progenitor cells when these differentiate into LCs in mice (Perea-Gomez et al., 2025).

That our organoids achieve gonadal marker expression without sustained *NR5A1* expression, albeit without the emergence of mature LCs, raises interesting questions about the precise threshold and timing of *NR5A1*/SF1 expression required for human fetal gonadal development. Indeed, it is possible that earlier significant *NR5A1* upregulation occurs within a small transient window during our differentiation protocol, which would only be captured with analysis over a detailed timeline. A similar phenomenon has been demonstrated for *SRY* expression in a testis-like cell differentiation protocol (Houzelstein et al., 2024). Our inducible PCS_AAVSI_SF1 cell line offers precise temporal control over SF1 expression, and future studies involving more frequent temporal sampling could help identify crucial developmental windows to optimise the timing of SF1 induction during organoid generation.

Taken together, our findings show that our testis-like organoids are a valuable model system for studying the molecular mechanisms governing DSDs that are caused by the disruption of genes that regulate the development of somatic gonadal cells. Notably, the Y chromosome independence of our system makes our model particularly valuable for studying DSDs where the typical SRY-initiated cascade may be disrupted. Indeed, our testis-like organoids show robust expression of the DSD genes curated in the PanelApp DSD gene list and we recently used this model to investigate DSD patient variants in the *SART3* (squamous cell carcinoma antigen recognised by T cells 3) gene (Ayers et al., 2023). Importantly, the application of both bulk RNA-seq and scRNA-seq analysis to our testis-like organoids has generated a rich dataset that will serve as a valuable resource for the field. Our findings contribute to the growing body of work on *in vitro* gonadal development and differentiation systems.

### Limitations of this study
A limitation of this study is the use of only endpoint measurements rather than intermediate time points during the differentiation protocol. Future studies would benefit from more frequent sampling intervals to provide a more comprehensive temporal analysis of the differentiation process and to better understand the molecular changes that occur throughout the protocol. We further note that, in our previous publication (Knarston et al., 2020), we described *SOX9*, *AMH* and *DHH* expression in organoids as upregulated when organoids were matured to 21 days, but we also found that the extent

of induction is variable between differentiations, a common challenge in organoid systems (Jensen and Little, 2023).

Of note, while cluster 2 was annotated as early SCs based on the expression of *SOX9* and other early SC markers (*BMPR1B*, *RARG*, *SIX4*, *COL2A1*), *GATA4* was only detected in a small subset of cells in this cluster. The incomplete co-expression of *SOX9* and *GATA4* might reflect either incomplete maturation of the induced SCs, or temporal dynamics where *GATA4* expression is asynchronous across differentiating cells. Future optimisation of our differentiation system may be required to achieve more homogenous expression of *SOX9* and *GATA4*.

Furthermore, while our data demonstrate upregulation of mature SC and LC markers following *NR5A1*/SF1 induction, we acknowledge that functional assays would be required to confirm SC functionality in the SF1-overexpressing organoids. However, such functional studies in testis organoid systems derived from hiPSCs remain technically challenging and are beyond the scope of this study. Future studies would not only benefit from developing a functional evaluation system for Sertoli-like cells in testis organoid systems, but also from a thorough evaluation of the factors in our differentiation medium that might facilitate *SRY*-independent masculinisation.

## MATERIALS AND METHODS
### Human iPSC culture and gonadal differentiation
PCS_201_010 iPS clone 5 hiPSCs (American Type Culture Collection) and PCS_AAVS1_SF1 hiPSCs were maintained at 5% $CO_2$ and 37°C in Essential 8 medium (E8; Thermo Fisher Scientific) with daily medium changes and passaged every 3-4 days using 0.5 mM EDTA/PBS (Thermo Fisher Scientific). Cell lines were routinely tested for contamination and authenticated. hiPSCs were differentiated into bipotential testis-like cells as described previously (Knarston et al., 2020). Briefly, dissociated cells were plated at 10,000 cells/cm² on 24-well plates coated with vitronectin (Thermo Fisher Scientific) in E8 with 1×RevitaCell (Thermo Fisher Scientific). After 24 h, medium was switched to Essential 6 (E6) with 3 μM CHIR99021 (R&D Systems) for 4 days. E6 was either home-made [500 ml DMEM/F12 medium (Thermo Fisher Scientific), 270 mg sodium bicarbonate, 7 mg sodium selenite, 32 mg L-ascorbic acid 2-phosphate sesquimagnesium salt hydrate, 5 mg Holo-Transferrin, and 10 mg insulin (all Sigma-Aldrich)], or phenol red-free E6 [same as home-made but with 500 ml phenol red-free DMEM/F12 (Thermo Fisher Scientific)]. From day 4 to day 7, E6 was supplemented with 200 ng/ml FGF9, 10 ng/ml BMP4 (both R&D Systems) and 1 μg/ml Heparin (Sigma-Aldrich). Testis-like organoids were generated on day 7 as described previously (Knarston et al., 2020; Takasato et al., 2015). Briefly, differentiated cells were dissociated with TrypLE Select (Thermo Fisher Scientific) for 2 min at 37°C, then 350,000 cells per organoid were pelleted (300 *g* for 3×3 min, with 180° rotations between spins). Pellets were transferred to Transwell filters (Corning) using wide bore pipette tips (Sigma-Aldrich). Then 1.2 ml of E6 was added underneath each Transwell filter, and media was changed every 2-3 days. From day 10 onwards, E6 was supplemented with 500 ng/ml PGD2 (Cayman Chemical). Organoids were harvested for analysis on differentiation days 18 and 21. To test additional small molecules and hormones in the gonadal differentiation protocol, 1 mM 8-Br-cAMP and 5 ng/ml LH, 200 mM SAG, 5 μM 22R-OHC, 5 mM LiCl (all Sigma-Aldrich) were added from differentiation day 10 onwards either alone or in combination. For DOX treatment of the DOX-inducible PCS_AAVS1_SF1 hiPSC line, DOX was added at a final concentration of 1 μg/ml from differentiation day 2 onwards.

### Generation of PCS_AAVS1_SF1 hiPSCs
The DOX-inducible AAVS1 donor plasmid (AAVS1-NGN2) was kindly provided by Dr Michael Peitz (University of Bonn, Germany) (Meijer et al., 2019). For the generation of the AAVS1-*NR5A1* construct, the *NGN2* (*NEUROG2*) gene was cut out from the donor vector by SalI and MluI digest and replaced with the *NR5A1* gene amplified from the pCMV6-Entry-hNR5A1 plasmid using the NEBuilder HiFi DNA Assembly Cloning Kit (New England Biolabs).

On the day before transfection, PCS_201_010 hiPSCs were dissociated into single cells with Accutase when the cells reached 70~80% confluence. Approximately $1\times10^5$ cells were seeded in each well of a vitronectin-coated 24-well plate in E8 medium supplemented with 10 µM Y-27632 (Hello Bio). After 24 h of incubation, the medium was changed with fresh E8 medium containing 10 µM Y-27632. Cells were transfected using Lipofectamine Stem (Thermo Fisher Scientific) with 500 ng TrueCut Cas9 protein (Thermo Fisher Scientific) complexed with 3 µl of 1 µM AAVS1 Alt-R sgRNA GGGGCCACTAGGGACAGGAT (Integrated DNA Technologies), 500 ng AAVS1-NR5A1 plasmid and 100 ng pCE-mp53DD plasmid (Addgene plasmid #41856). On the following day, cells were split with Accutase into a six-well plate. Transfected cells were selected with 0.3 µg/ml puromycin for a week and stable colonies were manually picked for clonal expansion. Genotyping was performed using Phire Tissue Direct PCR Master Mix (Thermo Fisher Scientific). Targeted insertions at the AAVS1 site were verified using two pairs of primers: primer pair 1 forward TGCTTTCTTTGCCTGGACAC, reverse GGTTCTGGCAAGGAGAGA-GA; primer pair 2 forward CCATAGCTCAGGTCTGGTCTAT, reverse AGGAAGAGAAGAGGTCAGAAGC.

### Embryoid body formation (pluripotency assay)
hiPSCs were plated on ultra-low attachment 96-well plates (Corning) in Essential 8 (Thermo Fisher Scientific) with 0.5% polyvinyl alcohol (PVA, Sigma-Aldrich) and 1×RevitaCell (Thermo Fisher Scientific). After 24 h, media was switched to Essential 8 without PVA and RevitaCell, and media was changed every 2-3 days. After two weeks of culture, embryoid bodies were dissociated with TrypLE Select (Thermo Fisher Scientific) for 2 min at 37°C and cells were plated on Matrigel-coated (Corning) chamber slides in Essential 8. Cells were cultured for another 2 weeks before fixing for immunofluorescence staining.

### RNA isolation and RT-qPCRs
RNA was extracted using the ReliaPrep RNA Cell Miniprep System (Promega) including DNase treatment (Promega) according to the manufacturer's instructions. cDNA was synthesised using the GoScript reverse transcriptase system (Promega). qRT-PCRs were performed using GoTaq qPCR Master Mix (Promega) on the LightCyclter480 (Roche) with *GAPDH* as reference gene. Primer sequences are listed in Table S3.

### Immunofluorescence (IF) staining
Cells and organoids were fixed with 4% paraformaldehyde in PBS for 15 min at room temperature, washed three times with PBS and kept in PBS at 4°C until further analysis. For whole mount IF staining, organoids were blocked for 2 h at room temperature [blocking buffer: 10% donkey serum, 5% bovine serum albumin in PBT-X (0.1% Triton X in PBS)]. Organoids were incubated with primary antibodies in 25% blocking buffer in PBT-X at 4°C for 72 h, followed by secondary antibodies in 25% blocking buffer in PBT-X for 2 h at room temperature. Nuclei were stained with DAPI (1:5000 in PBS). Organoids were mounted on glass cover slides using Fluoromount and imaged on a confocal microscope (LSM780, Zeiss). For sections, fixed organoids were allowed to settle in PBS/sucrose solution and then embedded in OCT, then 10 µm sections were mounted onto Superfrost Plus slides and stained as above. Antibody concentrations are listed in Table S4.

### Western blotting
Cells were lysed in 100 µl 2×SDS loading buffer with 10% DTT and denatured for 5 min at 95°C. Then, 20 µl protein lysate was transferred to PVDF membrane, blocked with 5% milk and incubated with primary antibodies overnight at 4°C, and with secondary antibody at room temperature for 2 h. Proteins were visualised using Amersham ECL Prime western blotting detection reagent (Cytiva) and the GE Amersham Imager 680. Antibody concentrations are listed in Table S4.

### Sample processing for bulk RNA-sequencing
For bulk RNA-seq, RNA from undifferentiated hiPSCs and organoids was isolated as described above. Library preparation and sequencing were carried out by VCGS (Victorian Clinical Genetic Services, Melbourne,

Australia). Samples were processed at VCGS using the Illumina stranded mRNA Library Prep kit. Sequencing was performed on a NovaSeq 6000 with 2×150 bp read length. Real-time image analysis was performed by NovaSeq Control Software (NCS) v1.7.5 in conjunction with Real Time Analysis (RTA) v3.4.4, running on the instrument computer. FASTQ generation was performed using Illumina bcl2fastq v2.20.0.422 pipeline.

### Sample processing for scRNA-sequencing
For scRNA-seq, organoids were harvested and dissociated to single cells using a modified enzymatic digestion protocol. Three organoids were pooled and manually chopped using scalpel blades in a small volume of media before treatment with 1 ml of Accutase:TrypLE (1:1) solution. Samples were incubated at 37°C for a maximum of 15 min with intermittent pipetting every 3-5 min to facilitate dissociation into single cells. The dissociation reaction was stopped by placing samples on ice, and the resulting cell suspension was filtered through a 40 µm membrane filter to remove debris and cell clumps. Cells were then washed, resuspended in PBS containing 0.04% bovine serum albumin, counted, and adjusted to a final concentration of 1000 cells/µl. Library preparation and sequencing were carried out by VCGS. Samples were processed at VCGS using the 10x Genomics Chromium Single Cell 3′ v3 kit with a targeted recovery of 10,000 cells per sample. Sequencing to 50,000 reads per cell was performed on an Illumina NovaSeq 6000 with a 28/8/91 single-indexed read configuration. CellRanger 3.1.0 was used for demultiplexing and FASTQ generation.

### Quantification and statistical analysis
#### Analysis of RT-qPCR data
Gene expression was normalised using the comparative threshold cycle (ΔΔCt) method. GAPDH was used as reference gene, and expression levels were calculated relative to undifferentiated samples at day 0. Statistical analysis of qRT-PCR data was performed using Python (scipy and scikit_posthocs). The distribution of gene expression data was first assessed for normality using the Shapiro-Wilk test. Based on the normality test results, for parametric distributed data comparisons were performed using one-way ANOVA followed by Dunnett's multiple comparison test. For non-parametric distributed data, comparisons were analysed using Kruskal–Wallis test followed by Dunn's multiple comparison test.

#### Bulk RNA-sequencing data processing and analysis
Raw RNA-seq data were processed and analysed using the Galaxy bioinformatics platform (Dobin et al., 2013). Raw sequencing reads underwent initial quality assessment using FastQC (0.11.9+galaxy 0). Adaptor sequences were trimmed using Cutadapt (Martin, 2011) (4.0+galaxy 0) and low-quality bases and short reads were removed using the following criteria: minimum read length (20 bps) and minimum base quality score (20). Processed reads were aligned to the built-in reference transcriptome GRCh38_107 using the STAR aligner (Dobin et al., 2013) (2.7.8a+galaxy 0) in two-pass mode. Duplicate reads were removed using picard (2.18.2.2) and gene expression quantification was performed using featurecounts (https://github.com/bgruening/galaxytools; 2.0.1+galaxy 2) with a minimum mapping quality per read of 10. Differential gene expression analysis was performed using the limma (Ritchie et al., 2015) package (3.60.4) through the Degust (Powell, 2019) web platform (degust.erc.monash.edu), which provides an interactive web platform for the exploration of RNA-seq results. The analysis included multiple testing correction (Benjamini-Hochberg FDR), and genes were considered significantly differentially expressed based on the following criteria: adjusted *P*-value (<0.05), log2 fold change (>1). Visualisation and further analysis of differential gene expression were conducted within the Degust platform.

#### Single cell RNA-sequencing data processing and analysis
scRNA-seq data were processed and analysed with R (4.4.2) using the Seurat (Hao et al., 2024) package (5.1.0). Raw sequencing data underwent initial quality control and preprocessing steps. Cells were filtered on the following criteria: minimum number of genes detected per cell (>200),

maximum percentage of mitochondrial genes (<10%) and number of genes per cell (>1500, <7000). Gene expression data were normalised using the Seurat LogNormalize function with a scale factor of 10,000. Cell cycle phase was determined using the Seurat CellCycleScoring function with canonical S and G2/M phase marker genes. Data were processed using SCTransform to regress out technical variation by accounting for mitochondrial percentage and cell cycle scores (G2/M and S phase). Highly variable genes in the dataset were then identified using the FindVariableFeatures function with default parameters. Principal component analysis (PCA) was performed, and the first two principal components were visualised. Uniform Manifold Approximation and Projection (UMAP) was used for dimensional reduction and visualisation. Clustering was performed using the FindClusters function with a resolution parameter of 0.2. Cluster identification was based on graph-based clustering algorithms in Seurat (Clustree 0.5.1). Cell types were annotated by examining marker gene expression and comparing to known cell type-specific markers using FindAllMarkers function. Data was handled and visualised using the dplyr (https://dplyr.tidyverse.org/) (1.1.4) and ggplot2 (Wickham, 2016) (2.3.5.1.) packages.

### Public dataset analysis and visualisation

To compare the expression of marker genes with a published dataset of human foetal gonads (Garcia-Alonso et al., 2022), we used the Reproductive Cell Atlas (https://www.reproductivecellatlas.org/gonads.html). Gene lists were queried through the cellxgene interface, and the resulting expression values were overlaid as colour intensity on the pre-existing UMAP coordinates. The colour coding in our feature plots represents the mean expression levels of the queried genes, as calculated and displayed by the cellxgene platform.

### Acknowledgements

The Novo Nordisk Foundation Center for Stem Cell Medicine, reNEW, is supported by a Novo Nordisk Foundation grant number (NNF21CC0073729). The Australian Regenerative Medicine Institute is supported by grants from the State Government of Victoria and the Australian Government.

### Competing interests

F.J.R. receives institutional and salary support as a coinvestigator and subcontractor with the Peter MacCallum Cancer Centre (Melbourne, Australia) for an investigator-initiated trial which receives funding support from Regeneron Pharmaceuticals, and as co-investigator on a translational research project funded by a Regeneron Pharmaceuticals grant.

### Author contributions

Conceptualization: S.P., M.H.L., A.H.S., K.L.A.; Data curation: S.P., N.C., J.T., D.B., S.B.W.; Formal analysis: S.P., G.R., N.C., E.M., S.B.W., K.L.A.; Funding acquisition: S.P., A.H.S., K.L.A.; Investigation: S.P., G.R., L.G.A.F., J.G., E.M., J.v.d.B., K.L.A.; Methodology: S.P., N.C., F.J.R., B.R., M.H.L., M.R., A.H.S., K.L.A.; Project administration: S.P., K.L.A.; Resources: A.H.S., K.L.A.; Supervision: S.P., F.J.R., B.R., M.H.L., M.R., A.H.S., K.L.A.; Validation: S.P., G.R., L.G.A.F., K.L.A.; Visualization: S.P., J.T.; Writing – original draft: S.P.; Writing – review & editing: S.P., G.R., M.H.L., M.R., A.H.S., K.L.A.

### Funding

This project was supported by an Ideas Grant (S.P., K.L.A.; APP2012250) from the National Health and Medical Research Council, Australia, by the National Stem Cell Foundation of Australia (S.P., K.L.A.), and by the Cybec Foundation (S.P., K.L.A.). Open Access funding provided by University of Melbourne. Deposited in PMC for immediate release.

### Data and resource availability

RNA-seq data reported in this paper have been deposited in GEO under accession number GSE285050. scRNA-seq data have been deposited in GEO under accession number GSE285051. The scripts used for scRNA-seq data processing are publicly available on GitHub at https://github.com/Ramialison-Lab/scRNAseq_workflow. All other relevant data and details of resources can be found within the article and its supplementary information.

### Peer review history

The peer review history is available online at https://journals.biologists.com/dev/lookup/doi/10.1242/dev.204772.reviewer-comments.pdf

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
