## [Peer Review File · Development (Cambridge, England)]

The emergence of multiple testicular cell lineages in human stem cell-derived testis-like organoids

Svenja Pachernegg, Gorjana Robevska, Lucas G.A. Ferreira, Natalie Charitakis, Jinchao Gu, Jan Terhag, Eliza Martin, Denis Bienroth, Jocelyn van den Bergen, Sean B. Wilson, Fernando J. Rossello, Ben Rollo, Melissa H. Little, Mirana Ramialison, Andrew H. Sinclair and Katie L. Ayers

DOI: 10.1242/dev.204772

Editor: Haruhiko Koseki

Review timeline

Original submission:	7 March 2025
Editorial decision:	15 April 2025
First revision received:	19 August 2025
Editorial decision:	17 September 2025
Second revision received:	16 December 2025
Editorial decision:	15 January 2026
Third revision received:	23 January 2026
Accepted:	27 January 2026

Original submission

First decision letter

MS ID#: dev.204772

MS TITLE: The emergence of multiple testicular cell lineages in human stem cell-derived testis-like organoids

AUTHORS: Svenja Pachernegg, Gorjana Robevska, Lucas G.A. Ferreira, Natalie Charitakis, Jinchao Gu, Jan Terhag, Eliza Martin, Denis Bienroth, Jocelyn van den Bergen, Sean B. Wilson, Fernando J. Rossello, Ben Rollo, Melissa H. Little, Mirana Ramialison, Andrew H. Sinclair and Katie L. Ayers

Dear Dr Pachernegg,

I have now received all the referees' reports on the above manuscript, and have reached a decision. The referees' comments are appended below, or you can access them online: please go to:

As you will see, the referees express considerable interest in your work, but have some significant criticisms and recommend a substantial revision of your manuscript before we can consider publication. If you are able to revise the manuscript along the lines suggested, which may involve further experiments, I will be happy receive a revised version of the manuscript. Your revised paper will be re-reviewed by one or more of the original referees, and acceptance of your manuscript will depend on your addressing satisfactorily the reviewers' major concerns. Please also note that Development will normally permit only one round of major revision. If it would be helpful, you are welcome to contact us to discuss your revision in greater detail. Please send us a point-by-point response indicating your plans for addressing the referees' comments, and we will look over this and provide further guidance.

Please attend to all of the reviewers' comments and ensure that you clearly highlight all changes made in the revised manuscript. Please avoid using 'Tracked changes' in Word files as these are lost in PDF conversion. I should be grateful if you would also provide a point-by-point response detailing how you have dealt with the points raised by the reviewers in the 'Response to Reviewers' box. If you do not agree with any of their criticisms or suggestions please explain clearly why this is so.

Reviewer 1

Advance summary and potential significance to field

The authors provide evidence that iPSC-derived cells can be differentiated into bipotential gonads and further into testicular cells (Sertoli and Leydig cells).

Comments for the author

- 1) The paper appears to present solid datasets. However, absence of intermediate timepoints weakens the validity. The plots comparing starting and final stages should not be combined as lines but need to be shown as bar charts as noting can be said for timepoints in between. In the previous paper such timepoints were presented and should also have been implemented here.
- 2) The order of results is surprising. In Figs. 2 and 3 almost no own data are shown. These could go into supplements. On the other hand Suppl. Fig. 3 contains highly relevant new data. These should be central datasets and should receive much more attention in results and discussion.
- 3) SOX9 data are confusing. Controls revealing SOX9 expression being specific for male development is missing. Female iPSC cells could have shown that ovarian development differs. Alternatively other markers could have been tested.

Reviewer 2

Advance summary and potential significance to field

This study explores human induced pluripotent stem cell (hiPSC)-derived testis-like organoids as a model for gonadal development and Differences of Sex Development (DSDs). Single-cell RNA transcriptome analysis identified six distinct cell clusters, including bipotential, early Sertoli, and testicular interstitial cell-like cell populations, each partially corresponding to previously published *in vivo* transcriptome profiles. To promote cell maturation, the authors overexpressed NR5A1 in an hiPSC line, successfully inducing Leydig cell marker expression. These findings provide a comprehensive transcriptional profile of hiPSC-derived testis-like organoids.

Although this paper presents a transcriptome analysis of the testicular organoids developed by the authors, it remains largely descriptive and lacks substantial biological insight. The authors tried to emphasize the similarity between the transcriptome profiles of their organoids and embryonic testicular cells, with a limited number of differentially expressed genes identified in each cluster. However, the extent to which this testis organoid system faithfully recapitulates the *in vivo* system remains unclear in this study. As pointed each critical point below, there are major issues to be clear. Given its descriptive nature and limited biological significance, the manuscript in its current form is not suitable for publication in Development.

1. The authors claim that their testicular organoid system can serve as a model for Differences of Sex Development (DSD). To support this, the transcriptome profile (including differentially expressed genes, DEGs) should be compared not only with embryonic testes but also with embryonic ovaries. Given that the transcriptome profile contains bipotential cells, some degree of overlap with embryonic ovarian transcriptomes is expected. Such a comparison would help clarify the specificity of male sex determination reproduced in this culture system.
2. Related to the comment above, it remains unclear whether this system relies on the presence of the Y chromosome. Would it be feasible to compare the transcriptome profiles of male

and female cells cultured under testicular differentiation conditions? This would provide insights into whether testicular differentiation in this system is inherently dependent on Y-linked factors.

3. For comparison to the transcriptomic profiles in vivo, a limited number DEGs based on Gene Ontology (GO) terms is not sufficient to support the authors' claims. Notably, aside from SOX9, few male-specific genes are mentioned. A more comprehensive analysis should include additional key male-determining genes such as DMRT1, AMH, and Sry. The authors should provide a more detailed characterization of the gene set involved in male sex determination.

4. The significance of NR5A1 overexpression remains unclear. While some downstream genes, such as AMH and DHH, are upregulated upon NR5A1 overexpression, it is difficult to determine whether these induced cells acquire Sertoli cell functionality or if NR5A1 overexpression simply leads to the induction of downstream genes. The authors should provide further evidence to distinguish between these possibilities.

Reviewer 3

Advance summary and potential significance to field

In this manuscript, based on the work by Knartson et. al. (2020), the authors analyzed gene expression in testis-like organoids derived from human iPS cells using both bulk and single-cell RNA sequencing. Through the analysis of the gene expression pattern and GO analysis of differentially expressed genes (DEGs), they hypothesized the existence of bipotential gonad cells and interstitial cells in addition to the Sertoli cell lineage. Single cell RNA-seq revealed several distinct clusters within the testis organoids, which the authors annotated using previously reported marker genes. Those induced cells lacked mature testis somatic cell markers, which led to the hypothesis that the forced expression of NR5A1 may restore the expression of such genes. To test this hypothesis, they generated testis organoid from a DOX-inducible NR5A1 iPSC cell line and demonstrated that NR5A1 over-expression up-regulates these mature markers through RT-qPCR experiment and IF analysis. These findings are interesting in that putative interstitial cells, supporting cells and steroidogenic cells are induced using a single methodology. However, I have some concerns regarding the interpretation of the data by the authors.

Major comments:

1. Even though cluster 2 in Fig. 3A highly expresses SOX9, these cells seem to lose the expression of WT1 and GATA4, as shown in Fig S3A. Furthermore, in the re-analysis of these data, cluster 2 expresses PDGFRA, which is an interstitial cell marker mentioned by the authors in this manuscript. Sertoli cells express WT1 and GATA4, but not PDGFRA in vivo. Since SOX9 is also expressed in some non-gonadal tissues, and WT1 and GATA4 are fundamental marker genes of gonad, I am skeptical of the authors claim that the cluster 2 represents the Sertoli cell lineage, and that it is a descendant of cluster 0 or 1.

Ensuring that male supporting cells are properly induced is an important factor in establishing an in vitro model of normal development and is therefore essential for its use as a disease model.

2. In the previous reports by Knartson et. al. (2020), both GATA4 and SOX9 were co-immunostained, and AMH were also stained. However, in the current manuscript, neither GATA4 nor AMH was expressed in the putative immature Sertoli lineage cells. Additionally, in Fig. 1C, the expression level of SOX9 does not differ between pre and post differentiation. In contrast, in previous reports, the expression level of SOX9 reaches a $\Delta\Delta CT$ value of 100 at day 21 in fig 6E of that study. Furthermore, in previous reports, Leydig cell markers, CYP11A1, CYP17A1, HSD3B and STAR were expressed without the forced expression of NR5A1. The author should address this discrepancy between the two studies.

3. Regarding the interstitial cells, there are two potential origins of the interstitial cells of testis in vivo: coelomic epithelium of gonad and mesonephros. The induced interstitial cells express WT1, GATA4 and ACTA2, suggesting that their origin is likely from gonadal cells. However, the expression levels of PDGFRA and TCF21 are weaker compared to those of the putative precursor cells. The authors should clarify the putative origin of the induced interstitial cells and compare the similarities and differences with the in vivo counterparts.

4. As the authors argued, supporting cells and interstitial cells are pre-existing cell populations, while steroidogenic cells appear to be newly emerged through forced expression of NR5A1. However, evidence for their existence is limited to bulk RNA expression data. Further immunohistochemical analysis or single cell RNA sequencing of the organoid could provide a clearer understanding of the marker gene expression and the origin of supporting cells, interstitial cells, and newly emerged steroidogenic cells. It would be important to evaluate and report in detail how the forced expression of NR5A1 improves this experimental system and better reflects *in vivo* development, highlighting the similarities and differences with *in vivo* development.

5. Regarding line 292-293: The up-regulation of AMH, DHH and FGF9 is indicative of Sertoli cell like differentiation, however, GATA4 and WT1 seems downregulated. The authors should clarify whether Sertoli cells are properly induced by NR5A1 forced expression, and these marker genes are expressed by Sertoli cells.

It is also the case for Leydig cell lineage cells and their markers such as STAR and CYP17A1. The author should check whether these genes are expressed by Leydig cell lineage cells or not. In addition, regarding Fig. 4B, there are many NR5A1 negative cells in this IF. Did the forced expression systems work well and is this iPS cell line derived from a single cell clone?

Minor comments:

Line 62: the authors state that "Stem cell technologies have opened new avenues for studying human development ...". However, the paper "Sato et al., 2009" does not contain data about human.

Line 133: PCA can only reduce dimensions up to the number of samples minus one. In other words, with only 3 or 4 samples, the PCA loadings can explain all the variance within 2 or 3 dimensions, making the statement "first two principal components explain most of the variance in both iPSC cells and organoids samples" in this manuscript an expected result, and therefore, there is no need to explicitly mention it. Additionally, what was the rationale for performing PCA within the sample groups?

Line 140: the authors state that various collagen genes and CLDN11 are expressed, however, it seems that there is no data about COL5A1, COL6A1, or CLDN11 in the cited articles in line 145 to 147.

I could find that all three genes were expressed in human testis data of Garcia-Alonso et al., 2022 in reprogonomics viewer. Please cite a correct literature, when necessary.

Line 151: Regarding the Fig. 1C heatmap, what is the unit of the color bar in the heatmap? In addition, the authors should clarify the process for creating cDNA and libraries for RNA seq in the Material and Methods section.

Line 167: Regarding Fig. 2A to C, it seems that some gene lists were used for public data analysis, but the details of how the gene list was used in the calculations to generate the UMAP projection are unclear. As a result, it is difficult to understand what the color coding represents. Could the authors please clarify how the gene list was used to calculate the expression levels and how those values were incorporated into the UMAP projection? Understanding the specific methodology behind these calculations would help in interpreting the meaning of these feature plots.

Line 218: regarding Fig. 3B, could the authors please clarify how the gene list was used to calculate the expression levels and how those values were incorporated into the UMAP projection.

Line 220: regarding RARG gene, there are no descriptions about RARG in the cited literature in line 222 to 223, but I can find some in "Leibich et al., 2022." Please cite a correct literature, when necessary.

Line 228-230: as mentioned in major comments 1 and 3, the authors should thoroughly evaluate the similarities and differences between *in vivo* and *in vitro* differentiation trajectory and marker gene expression. Currently, there is insufficient evidence to support the claim that *in vitro* differentiation aligns with that of *in vivo*.

Line 235: there is no Fig. 3D in this manuscript.

Line 308 and line 346-347: as mentioned above, please thoroughly compare the *in vivo* and *in vitro* data to identify and define the corresponding *in vivo* counterparts for each annotation.

Line 318: "[insulinexpression in LCs", it seems to be a typo.

Line 319-322: Since this part is a repetition of what is stated after line 313, it may not be necessary.

Line 333: regarding ACTA2, CALD1, DES, MYH11, it seems that there is no data about these genes in the cited literature in line 333 to 334. Please cite a correct literature, when necessary.

Line 353-356: as mentioned in major comments, please check these genes are expressed in putative Sertoli cells and putative Leydig cells.

Line 454-457: Table S1-S3 could not be evaluated by this reviewer.

Line 562: please disclose the antibodies and their concentrations for IF and Western blot. In addition, please provide the detailed protocol of Western blotting.

Line 601: please disclose the procedure of single cell RNA-seq sample preparation and library construction. In addition, if you used 10x chromium, please provide the version of chemistry and Cell Ranger.

Line 606: Is this threshold for "UMI counts" a typo? Should it be "the number of genes per cell," instead? In addition, have you considered the need for doublet removal?

First revision

Author response to reviewers' comments

Reviewer 1

1. The paper appears to present solid datasets. However, absence of intermediate timepoints weakens the validity. The plots comparing starting and final stages should not be combined as lines but need to be shown as bar charts as noting can be said for timepoints in between. In the previous paper such timepoints were presented and should also have been implemented here.

We thank the reviewer for this comment and agree that our representation of the data can be shown as bar charts. We have revised the figure (Fig. 5) to display the data as bar charts as suggested. We have not included intermediate time points as the goal of the research was to investigate conditions for a better endpoint for our differentiation protocol. We have added a note in the limitation's sections discussing how future work could benefit from more frequent sampling intervals to provide a more comprehensive temporal analysis.

2. The order of results is surprising. In Figs. 2 and 3 almost no own data are shown. These could go into supplements. On the other hand Suppl. Fig. 3 contains highly relevant new data. These should be central datasets and should receive much more attention in results and discussion.

We thank the reviewer for the feedback regarding the organisation of our results. We have moved the original Fig. 2 to the supplements (now Supp Fig. 2) and reorganised Fig. 3 by splitting it into a new Fig. 2 and Supp Fig. 3. Most importantly, we have promoted the original Supp Fig. 3, which contains novel data, to the main text as Figure 4 and have expanded its discussion in both the results and discussion sections. We have updated all figure numbering, references, and legends accordingly throughout the manuscript.

3. SOX9 data are confusing. Controls revealing SOX9 expression being specific for male development is missing. Female iPSCs could have shown that ovarian development differs. Alternatively other markers could have been tested.

We thank the reviewer for this question. We are not totally sure which dataset the reviewer is talking about here (bulk RNA seq or scRNA-seq), but we have attempted to address the concern about SOX9 below.

In our previous paper, we described SOX9 expression in organoids as upregulated when organoids were matured to 21 days, however had found that when this induction happens it is variable between differentiations. This is part of the reason we wanted to better characterise our organoids as we have done here.

Indeed, in our bulk RNA-seq data here we have sampled day 18 organoids - an earlier time point at which we hoped to see SOX9 becoming upregulated. In this data, capturing meaningful SOX9 expression is tricky as this gene is also expressed at medium level in all iPSCs meaning that when we compare its expression relative to day -1 (iPSCs) in our bulk RNA-seq, SOX9 does not appear to be

strongly upregulated. However, it is expressed, even slightly upregulated (but this varies between organoids). Protein levels for SOX9 do differ though, and compared to iPSCs, SOX9 protein is stronger in the organoids but in just a subset of cells (see our previous paper). Thus, it may be that its expression becomes more restricted to a smaller number of cells, which express it at a higher level. This is confirmed in our scRNA-seq data which shows multiple cell types in the organoids, only some of which express SOX9 strongly (Figs. 2, 3).

The inclusion of female iPSCs may not inform on this. We have previously demonstrated that SOX9 expression can occur in female iPSC lines in the absence of SRY (Knarston et al., 2020). We postulate this is due to the addition of FGF9 during days 4-7, which is a known regulator of SOX9 expression, and likely bi-passes the need for SRY induction of SOX9. Indeed, we do not see substantial SRY expression in the organoids at day 18 or 21 (consistent with other differentiation protocols from other groups who have shown that SRY expression peaks earlier, within the first 84 hours of a differentiation). We have added this explanation to the discussion. Nevertheless, to address the reviewer's concern about additional markers, we have included additional early gonadal or testis markers in our analysis including GADD45G, CBX2, EMX2, LHX9, ZFPM2/FOG2, and NROB1/DAX1. These additional marker analyses have been added as new Fig. 3 to provide a broader molecular context. We have updated the results and discussion accordingly.

Reviewer 2

1. The authors claim that their testicular organoid system can serve as a model for Differences of Sex Development (DSD). To support this, the transcriptome profile (including differentially expressed genes, DEGs) should be compared not only with embryonic testes but also with embryonic ovaries. Given that the transcriptome profile contains bipotential cells, some degree of overlap with embryonic ovarian transcriptomes is expected. Such a comparison would help clarify the specificity of male sex determination reproduced in this culture system.

We thank the reviewer for the suggestion to strengthen our DSD model validation by comparing our organoid transcriptome to both embryonic testes and ovaries. We have now performed this comparative analysis with embryonic ovaries and included the results as Supplementary Fig. 2A. As expected, we observed overlap with embryonic ovarian transcriptomes, consistent with the presence of bipotential cells within our system. This comparison has been expanded upon to clarify how our culture system recapitulates male sex determination pathways while also maintaining bipotential characteristics relevant for DSD modelling.

To this last point, we wanted to remind the reviewer that 46,XY DSD such as gonadal dysgenesis or 46,XX ovarian dysgenesis, may arise due to a disruption to early bi-potent testicular progenitors, and in many of these cases, testes or ovaries may never actually form, rather, patients can have non-functional fibrotic tissue (such as streak gonads in 46,XY complete gonadal dysgenesis). Therefore, some overlap with very early ovarian genes that also play a role in the bi-potential gonad can be desirable and overlap with known bi-potential signalling pathways is also important for DSD modelling.

2. Related to the comment above, it remains unclear whether this system relies on the presence of the Y chromosome. Would it be feasible to compare the transcriptome profiles of male and female cells cultured under testicular differentiation conditions? This would provide insights into whether testicular differentiation in this system is inherently dependent on Y-linked factors.

We thank the reviewer for this important question regarding Y chromosome dependency in our system. As shown in our previous publication (Knarston et al., 2020), our testicular differentiation protocol operates independently of the Y chromosome. Despite expression of bipotential gonad markers such as GADD45G and GATA4 that typically upregulate SRY, we failed to detect SRY RNA expression in our system at this later timepoint. Our approach bypasses the need for SRY by directly introducing FGF9 during days 4-7, which is a known activator of SOX9 and subsequently maintains testicular development through auto-regulation mechanisms involving factors like PGD2 (which is also added to the media from day 10 onwards). Crucially, we demonstrated that SOX9 expression and testicular differentiation occur even in female iPSC lines where SRY is absent, confirming that our system can achieve male gonadal development through the activation of downstream pathways

rather than relying on the Y chromosome. This Y chromosome independence makes our model particularly valuable for studying DSDs where the typical SRY-initiated cascade may be disrupted. We have added this explanation to the discussion. Interestingly, however, we have now included data showing the forced expression of NR5A1 does result in a small increase of SRY in organoids consistent with SF1 playing a role in SRY regulation (Fig. 5). We have also added this to the discussion.

3. For comparison to the transcriptomic profiles in vivo, a limited number DEGs based on Gene Ontology (GO) terms is not sufficient to support the authors' claims. Notably, aside from SOX9, few male-specific genes are mentioned. A more comprehensive analysis should include additional key male-determining genes such as DMRT1, AMH, and Sry. The authors should provide a more detailed characterization of the gene set involved in male sex determination.

We agree that a more detailed characterisation of male sex determination genes is important to support our claims. We have now conducted a comprehensive analysis of key male-determining genes including DMRT1, AMH, and SRY, which showed no expression in our system at these later organoid time point (which is consistent with Houzelstein et al 2024, where they showed SRY expression was only observed early in their testis differentiation protocol, before 84 hours). However, we detected expression of several other relevant genes including CBX2, GADD45G, EMX2, LHX9, ZFPM2/FOG2, and NROB1/DAX1. This expanded gene expression analysis has been added as a new Fig. 3 and provides a more complete molecular characterisation of our organoid system.

4. The significance of NR5A1 overexpression remains unclear. While some downstream genes, such as AMH and DHH, are upregulated upon NR5A1 overexpression, it is difficult to determine whether these induced cells acquire Sertoli cell functionality or if NR5A1 overexpression simply leads to the induction of downstream genes. The authors should provide further evidence to distinguish between these possibilities.

We thank the reviewer for raising this point about NR5A1 overexpression. We agree that distinguishing between simple gene induction and true Sertoli cell functionality is important. While we see upregulation of highly specific Sertoli cell markers like AMH and DHH following NR5A1 overexpression, we recognise that functional assays would be needed to confirm whether these cells actually behave as Sertoli cells. Unfortunately, testing Sertoli cell functionality in an organoid model is a challenge and not yet achieved by other groups using human iPSCs in a 3D system, meaning that currently our best tools to determine Sertoli cell differentiation are gene/protein expression of specific gene sets. We have now included analysis of additional genes (SRY, NR2F2, INHBB, HSD3B2 and NROB1) in this figure. Interestingly, markers of early bi-potential or precursor state, like EMX2, NR2F2, and GATA4 are either not changed or downregulated, whereas NROB1 and SRY (both known targets of SF1) show upregulation (Fig. 5). Whilst important, comprehensive functional studies are outside the current scope of this work but represent a key direction for future research, and we have added this to the Limitations of the study.

Reviewer 3

1. Even though cluster 2 in Fig. 3A highly expresses SOX9, these cells seem to lose the expression of WT1 and GATA4, as shown in Fig S3A. Furthermore, in the re-analysis of these data, cluster 2 expresses PDGFRA, which is an interstitial cell marker mentioned by the authors in this manuscript. Sertoli cells express WT1 and GATA4, but not PDGFRA in vivo. Since SOX9 is also expressed in some non-gonadal tissues, and WT1 and GATA4 are fundamental marker genes of gonad, I am skeptical of the authors claim that the cluster 2 represents the Sertoli cell lineage, and that it is a descendant of cluster 0 or 1. Ensuring that male supporting cells are properly induced is an important factor in establishing an in vitro model of normal development and is therefore essential for its use as a disease model.

We thank the reviewer for this comment, however we respectively disagree with several of their points. Firstly, there are very few, if any, truly specific markers for Sertoli cells. Yes, SOX9 expression can be associated with chondrogenesis, but WT1 is a well-known marker for fetal kidney podocytes and is also expressed in muscles, and GATA4 is well known to be also involved in cardiac development. This means that as for many other differentiation models we must rely on combinations of markers and the very few more specific genes such as AMH (for Sertoli cells) and steroidogenic enzymes for the Leydig cells.

The reviewer also states that “Sertoli cells express WT1 and GATA4, but not PDGFRA in vivo”. If one looks at fully mature Sertoli cells this may hold true, but if we look at human foetal gonad data from weeks 6-7, early stages where Sertoli cell differentiation may not be complete (Garcia-Alonso 2022, Fig. 2b), PDGFRA expression is observed in Sertoli cells (20% of cells with mean expression of about 0.25). This fits our conclusions that our organoid model is at an immature state. To revisit this question, we carried out subset clustering of SOX9-expressing cells and confirmed that GATA4 and WT1 are indeed co-expressed within this cell population (Fig. 3B). We have added this to the results.

Finally, the reviewer mentions that we claim that “cluster 2 is descendant of cluster 0 or 1”. In fact, this kind of analysis does not imply descent - we do not know the relationship between clusters from this analysis - and we have not tried to claim this. Our clustering analysis identifies transcriptional states present in our organoids.

2. In the previous reports by Knarston et. al. (2020), both GATA4 and SOX9 were co-immunostained, and AMH were also stained. However, in the current manuscript, neither GATA4 nor AMH was expressed in the putative immature Sertoli lineage cells. Additionally, in Fig. 1C, the expression level of SOX9 does not differ between pre and post differentiation. In contrast, in previous reports, the expression level of SOX9 reaches a $\Delta\Delta CT$ value of 100 at day 21 in fig 6E of that study. Furthermore, in previous reports, Leydig cell markers, CYP11A1, CYP17A1, HSD3B and STAR were expressed without the forced expression of NR5A1. The author should address this between the two studies.

We thank the reviewer for this comment and would like to clarify. We do observe co-expression and co-staining of GATA4 and SOX9 in our organoid model (please see new Fig. 3B and 6D). As per Knarston et al, the expression of these has never overlapped 100%, with GATA4 often showing more widespread expression than SOX9. We have now shown this more definitively in our subset analysis (Fig. 3B). We have addressed the expression of SOX9 in Figure 1 above, stating that we see expression but no upregulation from iPSCs, which we believe is due to this being an earlier timepoint than what was shown in Knarston et al. The same holds true for Leydig cell markers. Additionally, we have used slightly different differentiation media in the current paper than in the previous publication (E6 without phenol red in the current paper vs. E6 with phenol red in the previous paper). We have added this explanation to the Limitations of the Study.

For our NR5A1 expression experiment, we have sampled the organoids at day 18 again as we wanted to see if we could boost these markers earlier. Indeed, this is three days earlier and in the un-induced line, just HSD3B1 and 2 and STAR show some expression. This is interesting as this indicates at this earlier stage, the master regulator STAR is becoming expressed, and it is then highly induced by SF1 - meaning that boosting NR5A1 expression at this stage can lead to dramatic upregulation of both Sertoli and Leydig markers at an earlier timepoint in our differentiation protocol.

3. Regarding the interstitial cells, there are two potential origins of the interstitial cells of testis in vivo: coelomic epithelium of gonad and mesonephros. The induced interstitial cells express WT1, GATA4 and ACTA2, suggesting that their origin is likely from gonadal cells. However, the expression levels of PDGFRA and TCF21 are weaker compared to those of the putative precursor cells. The authors should clarify the putative origin of the induced interstitial cells and compare the similarities and differences with the in vivo counterparts.

If we understand correctly, the reviewer is asking if there is evidence that interstitial cells populations in our organoids arise from a more coelomic epithelial or mesonephros-like progenitor. Interstitial cells include peritubular cells, vascular endothelial cells, vascular smooth muscle and other perivascular cells, steroidogenic Leydig cells and their undifferentiated mesenchymal progenitors, and immune cells, which under normal conditions is mostly comprised of testicular macrophages. We assume the reviewer is referring here mostly to the Leydig and PTM cells and their origin. While we agree that studies in mice have convincingly shown interstitial cells likely originate from both CE (Wnt5a) or mesonephros (Rarb, Nestin), often using cell tracing with different markers, we are not familiar with any definitive work that demonstrates the same dual origin in humans. In fact, until recently molecular signatures of human fetal mesonephroi have not been closely investigated. But, recent single cell and spatial analysis in human foetal

samples has described different populations of rete testis especially that may indicate different origins, but these are a snapshot in time that rely on a subset of markers and do not definitively show a direct origin. Both the CE and mesonephros are mesodermal structures, and our differentiation protocol does indeed drive cells through an intermediate mesodermal progenitor stage. But we do not appear to have extra-gonadal populations because GATA2 (shown to be negative in gonadal stromal cells and positive in mesonephric/epididymal stromal cells) is not expressed in our organoids (see new Sup Fig 4). Nor do we have markers of mature mesonephros or podocytes in our organoids (LHX1/LIM1, NPHS1, NPHS2, PAX2), therefore it is unclear to us the relevance of this question - there is no mesonephros-like population from which interstitial cells could arise in this closed system. Interestingly, CE markers CALB2, LRRN4 and UPK3B are not expressed in our organoids with slight expression of KRT7, but we do see PAX8 CXCL14 and CXCR4 positive cells - suggesting that we have a “supporting cell population”. We have added a new figure (Supp Fig. 4) to address these questions and expanded the results accordingly.

4. As the authors argued, supporting cells and interstitial cells are pre-existing cell populations, while steroidogenic cells appear to be newly emerged through forced expression of NR5A1. However, evidence for their existence is limited to bulk RNA expression data. Further immunohistochemical analysis or single cell RNA sequencing of the organoid could provide a clearer understanding of the marker gene expression and the origin of supporting cells, interstitial cells, and newly emerged steroidogenic cells. It would be important to evaluate and report in detail how the forced expression of NR5A1 improves this experimental system and better reflects in vivo development, highlighting the similarities and differences with in vivo development.

We thank the reviewer for this comment. Unfortunately, we do not currently have the resources available to carry out single cell RNA sequencing of the doxycycline induced NR5A1 conditions. Contrary to the comment above, we do not think that the steroidogenic cells are “newly emerged” in these conditions. In fact, given that we are looking at just one time point it is not possible to draw conclusions on lineage. Rather, we hypothesise that forced expression of NR5A1 during the differentiation has allowed precursor cell populations to further mature earlier (we have analysed these organoids at day 18, earlier than our previous paper which was at day 21). We postulate this is because NR5A1 directly regulates the expression of key Leydig cell regulators like STAR which is significantly upregulated. To further address this, we have revisited this experiment and have included another Leydig specific marker (HSD3B2, Fig. 5) and have also included more staining for the induced organoids and larger images, where you can see the upregulation of STAR in more cells in the dox organoids (Fig. 6C). It is also interesting to note that NR2F2, a marker of interstitial progenitors which is down regulated when these differentiate into Leydig cells in mice (Perea-Gomez 2025) is downregulated too while Leydig specific markers become upregulated.

Additionally, we also saw evidence of more mature markers of Sertoli cells too (including AMH, DHH and INHBB), although the difference between non-induced and induced was less than seen for Leydig markers. Again, this may indicate that NR5A1 expression is a key requirement for earlier maturation of these cell types in this model, which is similar to what is known in vivo in humans, where loss of NR5A1 causes gonadal dysgenesis associated with a loss of gonadal supporting cells.

5. Regarding line 292-293: The up-regulation of AMH, DHH and FGF9 is indicative of Sertoli cell like differentiation, however, GATA4 and WT1 seems downregulated. The authors should clarify whether Sertoli cells are properly induced by NR5A1 forced expression, and these marker genes are expressed by Sertoli cells. It is also the case for Leydig cell lineage cells and their markers such as STAR and CYP17A1. The author should check whether these genes are expressed by Leydig cell lineage cells or not. In addition, regarding Fig. 4B, there are many NR5A1 negative cells in this IF. Did the forced expression systems work well and is this iPS cell line derived from a single cell clone?

We thank the reviewer for this observation. Yes, while we see induction of AMH, DHH and FGF9 from what was previously quite low levels, the differences in WT1 expression are not significant, and it is highly induced in both conditions. But the reviewer is correct that GATA4 levels are reduced in the NR5A1 forced expression cells compared to the controls. We are unsure what this means and still note that GATA4 expression is still very high in both conditions compared to undifferentiated stem cells. However, in IF one can see this high expression across the organoids in both conditions (new Fig. 6D).

We have now included more IF images where you can see the extent of GATA4 expression (new Fig. 6D). It does appear while there is a reduction in the qPCR, in the organoids there are patches of higher expression for GATA4 in the doxycycline induced, suggesting perhaps it just becomes more restricted to certain cells. We are not sure why there is a discrepancy between RNA and protein levels here.

Without a functional assay for Sertoli cells in this 3D model (see our comment above and the added Limitations to the study), the co-expression of SOX9, AMH, DHH, INHBB and even some SRY, suggests that these are Sertoli cells, and expression of STAR and multiple other steroidogenic genes suggests the emergence of Leydig cells. In humans STAR and CYP17A1 (as well as CYP11A1 and HSD3B1) are considered as specific markers of the Leydig cells. Their increased expression in the dox-induced condition is indicative that at least a portion of cells have differentiated to a Leydig cell type as mentioned above.

*We confirm that the system was from a single clone. We generated 6 different clones and tested two for expression. One had two insertions of the NR5A1 cassette at the AAVS1 locus, the other had just one inserted copy. Both showed robust expression of NR5A1 after dox addition - *8000x - 10000x compared to uninduced at RNA level after 36 hours in iPSCs. We proceeded with just one clone to organoids. In iPSCs we found the expression of SF1 protein was induced in all cells at a lower level. In organoids, there was low level induction throughout, with a subset of cells that expressed very high levels- hence imaging, where it is hard to see the lower level in all cells due to the high level in a subset - and the need to not oversaturate the image. The western blots (Fig. 4D) also demonstrate that there is a large increase in protein. We have now included more images to give the reader a better picture (new Fig. 6C).*

Minor comments:

Line 62: the authors state that "Stem cell technologies have opened new avenues for studying human development ...". However, the paper "Sato et al., 2009" does not contain data about human.

We thank the reviewer for this observation and have now included a corrected citation (Spence et al., 2011).

Line 133: PCA can only reduce dimensions up to the number of samples minus one. In other words, with only 3 or 4 samples, the PCA loadings can explain all the variance within 2 or 3 dimensions, making the statement "first two principal components explain most of the variance in both iPSC cells and organoids samples" in this manuscript an expected result, and therefore, there is no need to explicitly mention it. Additionally, what was the rationale for performing PCA within the sample groups?

We thank the reviewer for the observation and acknowledge that with our limited sample sizes (3-4 samples per group), the first two principal components will mathematically explain most variance. We performed PCA separately within each sample group as a control measure to assess technical reproducibility within each condition, rather than to demonstrate biological variance. We agree that the statement about variance explanation is mathematically expected given our sample sizes and have removed the statement from the results.

Line 140: the authors state that various collagen genes and CLDN11 are expressed, however, it seems that there is no data about COL5A1, COL6A1, or CLDN11 in the cited articles in line 145 to 147. I could find that all three genes were expressed in human testis data of Garcia-Alonso et al., 2022 in reprogenomics viewer. Please cite a correct literature, when necessary.

We thank the reviewer for this observation and have included Garcia-Alonso et al., 2022 as citation for collagens and CLDN11. We have also added Stammler et al., 2016 as citation for CLDN11 expression in Sertoli cells.

Line 151: Regarding the Fig. 1C heatmap, what is the unit of the color bar in the heatmap? In addition, the authors should clarify the process for creating cDNA and libraries for RNA seq in the Material and

Methods section.

We have added the unit for the colour bar ($\log_2(\text{CPM}+1)$) to Fig. 1C. We have also added a subsection in the Material and Methods section regarding the creating of cDNA and libraries for RNA sequencing.

Line 167: Regarding Fig. 2A to C, it seems that some gene lists were used for public data analysis, but the details of how the gene list was used in the calculations to generate the UMAP projection are unclear. As a result, it is difficult to understand what the color coding represents. Could the authors please clarify how the gene list was used to calculate the expression levels and how those values were incorporated into the UMAP projection? Understanding the specific methodology behind these calculations would help in interpreting the meaning of these feature plots.

We thank the reviewer for this question. To clarify, we used the pre-computed UMAP coordinates from the Reproductive Cell Atlas (<https://www.reproductivecellatlas.org/gonads/human-main-male>). Gene lists were queried through the cellxgene interface, and the resulting expression values were overlaid as colour intensity on the pre-existing UMAP coordinates. The colour coding in our feature plots represents the mean expression levels of the queried genes, as calculated and displayed by the cellxgene platform. We have added this explanation to the Methods section.

Line 218: regarding Fig. 3B, could the authors please clarify how the gene list was used to calculate the expression levels and how those values were incorporated into the UMAP projection.

Please see our reply to the previous comment.

Line 220: regarding RARG gene, there are no descriptions about RARG in the cited literature in line 222 to 223, but I can find some in "Leibich et al., 2022." Please cite a correct literature, when necessary.

We thank the reviewer for this observation and have included Liebich et al., 2022 as citation.

Line 228-230: as mentioned in major comments 1 and 3, the authors should thoroughly evaluate the similarities and differences between in vivo and in vitro differentiation trajectory and marker gene expression. Currently, there is insufficient evidence to support the claim that in vitro differentiation aligns with that of in vivo.

Please see our reply to major comments 1 and 3. We have now added new marker gene analysis (Fig. 3 and Fig. 4) to address these questions, and after careful consideration, we have retracted our statements in the discussion that our system can fully recapitulate in vivo development. However, we still believe that we do see the emergence of multiple testis cell types in our organoids.

Line 235: there is no Fig. 3D in this manuscript.

We apologise for the confusion, we meant to reference Fig. 3C, which is now Fig. 2B.

Line 308 and line 346-347: as mentioned above, please thoroughly compare the in vivo and in vitro data to identify and define the corresponding in vivo counterparts for each annotation.

Please see our reply to the comment above.

Line 318: "[insulin expression in LCs", it seems to be a typo.

We thank the reviewer for this observation and have corrected this typo.

Line 319-322: Since this part is a repetition of what is stated after line 313, it may not be necessary.

We thank the reviewer for this observation and have deleted this sentence as it was indeed a repetition of the previous statement.

Line 333: regarding ACTA2, CALD1, DES, MYH11, it seems that there is no data about these genes in the cited literature in line 333 to 334. Please cite a correct literature, when necessary.

We thank the reviewer for this observation and have included Liebich et al., 2022 as citation for these genes.

Line 353-356: as mentioned in major comments, please check these genes are expressed in putative Sertoli cells and putative Leydig cells.

Please see our reply to the comments above.

Line 454-457: Table S1-S3 could not be evaluated by this reviewer.

We have re-uploaded the Supplemental Tables as one PDF as requested by the editorial team, however, we note that this might not be the most convenient way to evaluate Tables S1 and S2.

Line 562: please disclose the antibodies and their concentrations for IF and Western blot. In addition, please provide the detailed protocol of Western blotting.

We have added a protocol for Western blotting to the Material and Methods section and included antibodies and their concentrations for IF and Western blotting in a new table (Table S4).

Line 601: please disclose the procedure of single cell RNA-seq sample preparation and library construction. In addition, if you used 10x chromium, please provide the version of chemistry and Cell Ranger.

We have added a subsection in the Material and Methods section regarding the scRNA-seq sample prep and library construction. We have also included the chemistry and Cell Ranger version.

Line 606: Is this threshold for "UMI counts" a typo? Should it be "the number of genes per cell," instead? In addition, have you considered the need for doublet removal?

We thank the reviewer for identifying this error and for the suggestion regarding doublet removal. The reviewer is correct in that this was an error in our description. The threshold (>1500, <7000) refers to the number of genes per cell (nFeature_RNA), not UMI counts. We have corrected the methods text accordingly. Following the reviewer's suggestion, we also performed doublet detection using scDbtFinder on our processed dataset. We identified 58 predicted doublets (1.2%) out of 4,928 total cells. This low doublet rate is within the expected range for 10X Chromium technology and suggests minimal impact on our conclusions.

Second decision letter

MS ID#: dev.204772R1

MS TITLE: The emergence of multiple testicular cell lineages in human stem cell-derived testis-like organoids

AUTHORS: Svenja Pachernegg, Gorjana Robevska, Lucas G.A. Ferreira, Natalie Charitakis, Jinchao Gu, Jan Terhag, Eliza Martin, Denis Bienroth, Jocelyn van den Bergen, Sean B. Wilson, Fernando J. Rossello, Ben Rollo, Melissa H. Little, Mirana Ramialison, Andrew H. Sinclair and Katie L. Ayers

Dear Dr Pachernegg,

I have now received all the referees' reports on the above manuscript, and have reached a decision. The referees' comments are appended below, or you can access them online: please go to:

As you will see, the referees appreciate to your substantial efforts to revise the manuscript, but still have some significant concerns and recommend further revision of your manuscript before we can consider publication. If you are able to revise the manuscript along the lines suggested, which may involve further experiments, I will be happy receive a revised version of the manuscript. Your revised paper will be re-reviewed by one or more of the original referees, and acceptance of your manuscript will depend on your addressing satisfactorily the reviewers' major concerns. Please also note that Development will normally permit only one round of major revision. If it would be helpful, you are welcome to contact us to discuss your revision in greater detail. Please send us a point-by-point response indicating your plans for addressing the referees' comments, and we will look over this and provide further guidance.

Please attend to all of the reviewers' comments and ensure that you upload both a 'clean' version of your Word file, along with a highlighted version clearly showing where you have made changes in the revised manuscript. Please avoid using 'Tracked changes' in Word files as these are lost in PDF conversion. I should be grateful if you would also provide a point-by-point response detailing how you have dealt with the points raised by the reviewers in the 'Response to Reviewers' box. If you do not agree with any of their criticisms or suggestions please explain clearly why this is so.

Reviewer 1

Advance summary and potential significance to field

Sexual differentiation of gonads is a highly exciting and rapidly progressing field. This paper adds very relevant insights to the research area by using organoid models and determining many endpoints in cellular progression. The authors have responded appropriately to the many comments and suggestions by the reviewers and implemented all necessary changes. Specifically new data and better figures provide better and more complete insights.

Comments for the author

The authors created an excellent response to the criticism of the reviewers. The paper contains valid datasets and adds relevant information to the field of gonadal differentiation.

Reviewer 2

Advance summary and potential significance to field

The authors have made substantial revisions in response to the concerns raised. The additional analyses and expanded discussion not only strengthen the manuscript but also enhance its relevance for understanding testicular organoid systems. Although it remains unclear whether the system fully recapitulates testicular differentiation "particularly with respect to Y-chromosome dependency" the authors provide a plausible explanation that multiple pathways may contribute to sex determination in human gonads. This culture system may serve as a valuable candidate model for human gonadal sex determination, particularly in conjunction with other emerging systems. While further validation will be required, the present work is suitable for publication as a candidate model.

Followings are comments for each point raised at the first round of the review.

1. I appreciate the authors' effort to incorporate transcriptome comparisons with embryonic ovaries in addition to embryonic testes. The inclusion of Supplementary Fig. 2A supports the notion that the system contains bipotential cells while still recapitulating male sex determination pathways. Although sex-specific pathways are not yet entirely clear, this seems feasible given that the differentiation pathways of supporting cells are similar between males and females.

2. The explanation regarding Y chromosome independence, with reference to the prior study (Knarston et al., 2020), is understandable. The clarification that testicular differentiation is driven by FGF9 and SOX9 activation, even in female iPSC lines, may provide an insight into SRY-independent masculinization pathway in human gonads. The Y chromosome-independent

masculinization in the culture system may be triggered by factors in the culture medium, which warrants further evaluation in future studies.

3. The expanded analysis including CBX2, GADD45G, EMX2, LHX9, ZFPM2/FOG2, and NR0B1/DAX1 provides a more comprehensive molecular profile. Although the expression of several key genes such as AMH and DMRT1 is not clearly detected, this may reflect low expression levels and/or technical limitations of single-cell transcriptome analysis.

4. I commend the authors for acknowledging the limitations of inferring Sertoli cell functionality based solely on gene/protein expression. The expanded analysis of additional markers, along with the observation that precursor state markers are downregulated, provides a stronger case for NR5A1's role in promoting testicular differentiation. I also agree with the notion that there is currently no appropriate platform for validating the functionality of human Sertoli cells. But, future efforts should focus on developing an experimental system for validating the Sertoli-like cells obtained in this culture system.

Reviewer 3

Advance summary and potential significance to field

Major comment 1

The authors have provided some information that some of the marker genes of Sertoli cells are not so specific, and PDGFRA is expressed in early Sertoli cells in vivo. I accept the author's explanation about PDGFRA expression in Sertoli cells, given that it is consistent with previously published findings.

However, the analysis of Fig. 3B and the cluster 2 population in Fig. 2A are somewhat confusing. The feature plots of Fig. 3B may show SOX9-expressing cells using subset function ($SOX9 > 0.5$?), and it is true that some of them express GATA4 and WT1. But most of the cluster 2 cells (the majority of SOX9 positive cells), which are annotated as Early Sertoli cells, seem GATA4 negative as shown below. In general, Sertoli cells express both SOX9 and GATA4, as shown in Fig. 6D (without DOX). If the cluster 2 cells are Sertoli cells, which population was stained in Fig. 6D? Some comments or additional explanation may be helpful to understand this inconsistency.

(Figures uploaded as the Reviewer Attachment)

Major comment 2

I acknowledge the author's possible explanations regarding SOX9 expression of bulk RNA-seq before and after differentiation, and the forced expression of SF1 led to more fast expression of some Sertoli and Leydig markers than previously reported.

Major comment 3

I acknowledge the author's answer that the knowledge on the origin of human gonad is limited until now and appreciate their efforts to clarify the origin of induced somatic cells. As the authors mentioned, mesonephric like cells may not exist, or at least, they are the minor population in this induction protocol. Certainly, CE markers would be helpful to understand how these testis somatic cells were differentiated, but I understand that it is difficult to detect these genes in this dataset.

Regarding PAX8, CXCL14, and CXCR4 expression, these are all the marker genes of sPAX8, which are reported in Garcia-Alonso, et al, 2020. Do the authors assume the existence of sPAX8 like population in this system in Line 260-261? If that is the case, please cite the paper again here, and it would be helpful if the authors could provide some other interpretation of these marker genes-positive cells, and the expression pattern of PAX8 in this dataset, because the ubiquitous expression of PAX8 in Fig. S4 is unexpected.

Major comment 4

The authors have provided some explanation regarding the presence of Leydig cell lineage without DOX as they previously reported in Knartson, et al., 2020., which induced SOX9 and AMH positive

Sertoli like cells and HSD3B expressing steroidogenic cells. In that context, forced expression of NR5A1 possibly accelerated the differentiation or the marker gene expression of these population.

Major comment 5 and minor comment for Line 353-356 of the previous version.

I accept the authors explanation that the level of forced SF1 expression tends to be heterogenous even though the cell lines are derived from a single clone, and the expression of GATA4 is reduced by some unknown reasons. And Fig. 5 showed upregulation of some Sertoli and Leydig cell marker genes and indicated their differentiation. However, their immunostaining was limited to the simultaneous staining of DHH and STAR in Fig. 6C. That can prove that DHH positive cells and STAR positive cells are different populations. As Knartson previously showed, AMH and WT1 positive putative Sertoli like cells, and SOX9 negative, GATA4 and STAR positive putative Leydig like cells are induced in this system, but there is no robust evidence that DHH is expressed by Sertoli cell like population, and other Leydig cell markers are expressed by Leydig cells in this system. The authors might consider multiplexed immunostaining of DHH and some of Sertoli cell marker genes.

Minor comments

I appreciate the authors effort to address many minor comments and revise the manuscript. I have some additional minor comments as follows:

Line 52 and 70

Should "testis" be "testes"?

Line 90

Liang et al., 2019, the authors checked Leydig cell markers of AMH-EGFP positive cells and found that Leydig cell marker genes are not expressed or expressed at low level. Finally, the authors concluded that they induced only Sertoli cells.

Line 152

LGR5 seems to be expressed in ESGC of both sexes (Garcia-Alonso, et al., 2020). For this reason, LGR5 may not be an ovary specific marker gene in human.

Line 170

Is Table S2 true? It seems that Table S1 is correct.

Line 499

C) seems a typo of B). In addition, please write a criterion of SOX9 expression, or renew the code in Git hub for analysis of Fig. 3B.

Line 583

Is 180°C a typo?

Fig. 1E

The cpm in these graphs should be CPM.

Fig. S2A, B and legend for Table S1 at Supplementary information section

Are the words DGEs in the title of Top 500DGEs in FigS2A and the words DGEs in the color bars and legend for Table S1 correct? At Line 174 and 179, they are called as DEGs.

Second revision

Author response to reviewers' comments

Reviewer 1

1. The authors created an excellent response to the criticism of the reviewers. The paper contains

valid datasets and adds relevant information to the field of gonadal differentiation.

We sincerely thank the editor and all reviewers for their constructive evaluation of our manuscript. We appreciate the reviewer's positive assessment of our datasets and contributions to the field of human gonadal differentiation.

Reviewer 2

1. I appreciate the authors' effort to incorporate transcriptome comparisons with embryonic ovaries in addition to embryonic testes. The inclusion of Supplementary Fig. 2A supports the notion that the system contains bipotential cells while still recapitulating male sex determination pathways. Although sex-specific pathways are not yet entirely clear, this seems feasible given that the differentiation pathways of supporting cells are similar between males and females.

We thank the reviewer for their positive feedback regarding the inclusion of transcriptome comparisons with embryonic ovaries.

2. The explanation regarding Y chromosome independence, with reference to the prior study (Knarston et al., 2020), is understandable. The clarification that testicular differentiation is driven by FGF9 and SOX9 activation, even in female iPSC lines, may provide an insight into SRY-independent masculinization pathway in human gonads. The Y chromosome-independent masculinization in the culture system may be triggered by factors in the culture medium, which warrants further evaluation in future studies.

We thank the reviewer for their appreciation of our explanation regarding Y chromosome independence in our differentiation system. The reviewer raises an excellent point that culture medium factors may contribute to Y chromosome-independent masculinisation in our system. We acknowledge this possibility and agree that identifying such factors are an important direction for future studies. We have noted this in the Limitations of the study.

3. The expanded analysis including CBX2, GADD45G, EMX2, LHX9, ZFPM2/FOG2, and NR0B1/DAX1 provides a more comprehensive molecular profile. Although the expression of several key genes such as AMH and DMRT1 is not clearly detected, this may reflect low expression levels and/or technical limitations of single-cell transcriptome analysis.

We thank the reviewer for recognising the value of our expanded analysis, which provides a more comprehensive molecular profile of gonadal differentiation. We agree with the reviewer's assessment that the lack of clear detection of AMH and DMRT1 likely reflects their low expression levels in our system and/or technical limitations inherent to single cell RNA seq analysis, particularly for genes expressed at modest levels.

4. I commend the authors for acknowledging the limitations of inferring Sertoli cell functionality based solely on gene/protein expression. The expanded analysis of additional markers, along with the observation that precursor state markers are downregulated, provides a stronger case for NR5A1's role in promoting testicular differentiation. I also agree with the notion that there is currently no appropriate platform for validating the functionality of human Sertoli cells. But, future efforts should focus on developing an experimental system for validating the Sertoli-like cells obtained in this culture system.

We thank the reviewer for their positive feedback regarding our acknowledgment of the limitations of inferring Sertoli cell function from gene or protein expression alone, and that validating human Sertoli cell functionality remains a significant challenge in the field. We agree that developing experimental systems to functionally validate Sertoli-like cells are an important direction for future studies, and have noted this in Limitations of this study.

Reviewer 3

1. The authors have provided some information that some of the marker genes of Sertoli cells are not so specific, and PDGFRA is expressed in early Sertoli cells in vivo. I accept the author's explanation

about PDGFRA expression in Sertoli cells, given that it is consistent with previously published findings.

However, the analysis of Fig. 3B and the cluster 2 population in Fig. 2A are somewhat confusing. The feature plots of Fig. 3B may show SOX9-expressing cells using subset function (SOX9 > 0.5 ?), and it is true that some of them express GATA4 and WT1. But most of the cluster 2 cells (the majority of SOX9 positive cells), which are annotated as Early Sertoli cells, seem GATA4 negative as shown below. In general, Sertoli cells express both SOX9 and GATA4, as shown in Fig. 6D (without DOX). If the cluster 2 cells are Sertoli cells, which population was stained in Fig. 6D? Some comments or additional explanation may be helpful to understand this inconsistency.

We thank the reviewer for this follow-up comment regarding the apparent discrepancy between GATA4 expression patterns in cluster 2 (early Sertoli cells) in Fig. 2A/3B versus Fig. 6D. The feature plots in Fig. 3B were generated using cells with SOX9 expression >1 (log2-normalised counts) to highlight cells with robust SOX9 expression. We acknowledge that this is a relatively stringent cutoff and have now used an expression threshold of >0.5 log2-normalised counts. We have also performed the reverse analysis and looked at SOX9 (and WT1) expression in GATA4-expressing cells, and SOX9 (and GATA4) expression in WT1-expressing cells (the three panels are now Fig. S4A, B, C). We have also included the expression thresholds in the manuscript.

We see co-expression of all markers in some cells, but not all. Similarly, analysis of human fetal testis development by Guo et al. (2021) showed that at 6-7 weeks post-fertilisation, GATA4 and SOX9 expression show only partial overlap, similar to what we see in our scRNA data set (please see attached screenshot of SOX9 (left) and GATA4 (right) expression in the Guo dataset (<https://rgv.genouest.org/dataset/d1357>). This explanation has now been included in the Discussion.

NOTE: Figure provided for reviewer has been removed. It showed a figure from Guo J, Sosa E, Chitiashvili T, Nie X, Rojas EJ, Oliver E; DonorConnect; Plath K, Hotaling JM, Stukenborg JB, Clark AT, Cairns BR. Single-cell analysis of the developing human testis reveals somatic niche cell specification and fetal germline stem cell establishment. Cell Stem Cell. 2021 Apr 1;28(4):764-778.e4. doi: 10.1016/j.stem.2020.12.004. Epub 2021 Jan 15. PMID: 33453151; PMCID: PMC8026516. We have removed unpublished data that had been provided for the referees in confidence.

Regarding Fig. 6D (now Fig. S6), the immunostaining captures cells at an earlier time point (day 18 vs. day 21 in the scRNA seq analysis) and shows SOX9 and GATA4 co-expression in a subset of cells, but not in all cells. Further analysis in the dox-induced state has shown that SOX9 subcellular localisation may also play a role, with only a subset of GATA4 cells expressing SOX9, and only a subset of these, often those with nuclear localisation, expressing DHH. See comment 5 below.

2. I acknowledge the author's possible explanations regarding SOX9 expression of bulk RNA-seq before and after differentiation, and the forced expression of SF1 led to more fast expression of some Sertoli and Leydig markers than previously reported.

We appreciate the reviewer's acknowledgment that forced SF1 expression accelerates the expression of Sertoli and Leydig cells markers. This observation emphasises the usefulness of our approach for studying early events in gonadal differentiation.

3. I acknowledge the author's answer that the knowledge on the origin of human gonad is limited until now and appreciate their efforts to clarify the origin of induced somatic cells. As the authors mentioned, mesonephric like cells may not exist, or at least, they are the minor population in this induction protocol. Certainly, CE markers would be helpful to understand how these testis somatic cells were differentiated, but I understand that it is difficult to detect these genes in this dataset.

Regarding PAX8, CXCL14, and CXCR4 expression, these are all the marker genes of sPAX8, which are reported in Garcia-Alonso, et al, 2020. Do the authors assume the existence of sPAX8 like population in this system in Line 260-261? If that is the case, please cite the paper again here, and it would be helpful if the authors could provide some other interpretation of these marker

genes-positive cells, and the expression pattern of PAX8 in this dataset, because the ubiquitous expression of PAX8 in Fig. S4 is unexpected.

We thank the reviewer for acknowledging our efforts to clarify the origin of our induced somatic cells and for their understanding regarding the technical difficulties in detecting CE markers in our dataset. We looked at CALB2, UP3KB, LRRN4, KRT7 and saw no or only minimal expression of these markers (Fig. S4D).

Regarding PAX8, CXCL14, and CXCR4 expression, we appreciate the reviewer pointing out that these are marker genes of the sPAX8 population described in Garcia-Alonso et al. We have now added the citation where we discuss these markers.

Regarding our interpretation of PAX8 expression, while it is possible there is an sPAX8 like population, we do not necessarily assume this. Indeed, there may be a simpler explanation - the widespread expression of PAX8 in our dataset (Fig. S4D) may also reflect the intermediate mesodermal origin of our cells, as PAX8 is expressed early in the intermediate mesoderm (Bouchard et al., 2002). We have now added these ideas to our manuscript.

4. The authors have provided some explanation regarding the presence of Leydig cell lineage without DOX as they previously reported in Knartson, et al., 2020., which induced SOX9 and AMH positive Sertoli like cells and HSD3B expressing steroidogenic cells. In that context, forced expression of NR5A1 possibly accelerated the differentiation or the marker gene expression of these population.

We appreciate the reviewer's acknowledgment that forced SF1 expression accelerates the expression of Sertoli and Leydig cell markers.

5. and minor comment for Line 353-356 of the previous version. I accept the authors explanation that the level of forced SF1 expression tends to be heterogenous even though the cell lines are derived from a single clone, and the expression of GATA4 is reduced by some unknown reasons. And Fig. 5 showed upregulation of some Sertoli and Leydig cell marker genes and indicated their differentiation. However, their immunostaining was limited to the simultaneous staining of DHH and STAR in Fig. 6C. That can prove that DHH positive cells and STAR positive cells are different populations. As Knartson previously showed, AMH and WT1 positive putative Sertoli like cells, and SOX9 negative, GATA4 and STAR positive putative Leydig like cells are induced in this system, but there is no robust evidence that DHH is expressed by Sertoli cell like population, and other Leydig cell markers are expressed by Leydig cells in this system. The authors might consider multiplexed immunostaining of DHH and some of Sertoli cell marker genes.

We thank the reviewer for their comment and for acknowledging our explanation regarding the heterogeneity of NR5A1/SF1 expression and the observed reduction in GATA4 levels. We appreciate the reviewer's acknowledgement that our data in Fig. 5 demonstrate upregulation of Sertoli and Leydig cell markers.

In line with the reviewer's comments, we have carried out additional staining on the dox- induced organoids. This co-staining with SOX9, GATA4 and DHH has revealed that DHH expressing cells have high levels of SOX9 and GATA4. However, this analysis has also revealed an interesting observation - that cells expressing SOX9 at the highest levels show differences in subcellular localisation - where some show predominantly nuclear SOX9 whereas others show cytoplasmic localisation, something that has been observed before (Stewart et al., 2020). Of note, it appears that the cells with predominantly nuclear SOX9 are the cells that express DHH. Nevertheless, this additional staining suggests that DHH is expressed in cells with SOX9, and that these are representative of a Sertoli cell like population as opposed to a Leydig or additional population.

We have included examples of this in Figure 6D, and have included it in the results and discussion. We have moved some of the whole mount images from Figure 6 to a supplementary figure 6.

Minor comments:

Line 52 and 70: Should "testis" be "testes"?

We thank the reviewer for this comment and have corrected this typo.

Line 90: Lian et al., 2019, the authors checked Leydig cell markers of AMH-EGFP positive cells and found that Leydig cell marker genes are not expressed or expressed at low level. Finally, the authors concluded that they induced only Sertoli cells.

We thank the reviewer for this correction and have removed the citation.

Line 152: LGR5 seems to be expressed in ESGC of both sexes (Garcia-Alonso, et al., 2020). For this reason, LGR5 may not be an ovary specific marker gene in human.

We agree that this human dataset shows only patchy expression of LGR5 in both sexes and so it might not be a good marker for granulosa cells. We have changed the text and Fig. 1C to remove LGR5 from our analysis. It is extremely difficult to find granulosa markers from early fetal timepoints that are not also expressed or required early in the bipotential gonad. Indeed, RSPO1 is also known to play a role in the bipotential gonad, at least in mice, meaning that FOXL2 is the main markers we can use to distinguish granulosa cells. FOXL2 was not expressed in our cells.

Line 170: Is Table S2 true? It seems that Table S1 is correct.

We thank the reviewer for this comment and have corrected this typo.

Line 499: C) seems a typo of B). In addition, please write a criterion of SOX9 expression, or renew the code in Git hub for analysis of Fig. 3B.

We thank the reviewer for this comment and have included the threshold for SOX9 expression.

Line 583: Is 180°C a typo?

We thank the reviewer for this comment and have corrected this typo.

Fig .1E: The cpm in these graphs should be CPM.

We thank the reviewer for this comment and have corrected this typo.

Fig. S2A, B and legend for Table S1 at Supplementary information section: Are the words DGEs in the title of Top 500DGEs in Fig S2A and the words DGEs in the color bars and legend for Table S1 correct? At Line 174 and 179, they are called as DEGs.

We thank the reviewer for this comment and have corrected these - it should be DEGs in all cases.

Third decision letter

MS ID#: dev.204772R2

MS TITLE: The emergence of multiple testicular cell lineages in human stem cell-derived testis-like organoids

AUTHORS: Svenja Pachernegg, Gorjana Robevska, Lucas G.A. Ferreira, Natalie Charitakis, Jinchao Gu, Jan Terhag, Eliza Martin, Denis Bienroth, Jocelyn van den Bergen, Sean B. Wilson, Fernando J. Rossello, Ben Rollo, Melissa H. Little, Mirana Ramialison, Andrew H. Sinclair and Katie L. Ayers

Dear Dr Pachernegg,

I have now received all the referees reports on the above manuscript, and have reached a decision. The referees' comments are appended below.

The overall evaluation is positive and we would like to publish a revised manuscript in Development, provided that the referees' comments can be satisfactorily addressed. Please attend to all of the reviewers' comments in your revised manuscript and detail them in your point-by-point response. If you do not agree with any of their criticisms or suggestions explain clearly why this is so. If it would be helpful, you are welcome to contact us to discuss your revision in greater detail. Please send us a point-by-point response indicating your plans for addressing the referees' comments, and we will look over this and provide further guidance.

Reviewer 3

Advance summary and potential significance to field

Comment 1: I thank the authors for their response to this comment. My interpretation is that the partial overlap of SOX9 and GATA4 in Guo's datasets reflects the presence of Sertoli cells and GATA4-expressing, SOX9-negative interstitial cells, rather than true heterogeneity within Sertoli cells. In the author's dataset, the cluster 2 population broadly expresses SOX9, whereas GATA4 is detected in only a small subset of cells. The authors should therefore acknowledge this as a limitation of the induction system or the analysis, that only a subset of the cluster 2 population expresses GATA4, despite the cluster being annotated as early Sertoli cells.

Comment 3: I agree with the authors that, given the widespread expression of both PAX8 and WT1, it is reasonable to infer an intermediate mesodermal origin for the gonadal somatic cell-like population in this system. The interpretation may be strengthened by explicitly referring to the combined expression of PAX8 and WT1, rather than PAX8 alone.

Comment 5: I appreciate the authors' efforts in performing co-staining for GATA4, SOX9, and DHH. The colocalization of these proteins suggests that mature Sertoli cell-like cell states are induced by forced expression of NR5A1, a key driver of gonadal somatic cell differentiation.

minor comment

Line90: The citation of Liang et al., 2019. is not removed here.

FigS3B, line 187: SCEA seems a typo of CSEA.

line 134: The word DGE still remains here. Do the authors deliberately use the term DGE here, rather than DEG?

line 437: sPAX9 may be a typo of sPAX8.

Third revision

Author response to reviewers' comments

Reviewer 3

1. I thank the authors for their response to this comment. My interpretation is that the partial overlap of SOX9 and GATA4 in Guo's datasets reflects the presence of Sertoli cells and GATA4-expressing, SOX9-negative interstitial cells, rather than true heterogeneity within Sertoli cells. In the author's dataset, the cluster 2 population broadly expresses SOX9, whereas GATA4 is detected in only a small subset of cells. The authors should therefore acknowledge this as a limitation of the induction system or the analysis, that only a subset of the cluster 2 population expresses GATA4,

despite the cluster being annotated as early Sertoli cells.

We thank the reviewer for this clarification. We now acknowledge in the Limitations of the Study section, that GATA4 expression is detected in only a subset of cluster 2 cells, despite this cluster's annotation as early Sertoli cells based on SOX9 and other Sertoli cell marker expression. This heterogeneity likely reflects either incomplete maturation of the induced Sertoli cell population or technical limitations capturing asynchronous GATA4 expression during differentiation, and further optimisation of our differentiation system maybe required to achieve more homogenous expression of SOX9 and GATA4.

2. I agree with the authors that, given the widespread expression of both PAX8 and WT1, it is reasonable to infer an intermediate mesodermal origin for the gonadal somatic cell- like population in this system. The interpretation may be strengthened by explicitly referring to the combined expression of PAX8 and WT1, rather than PAX8 alone.

We thank the reviewer for this suggestion. We have revised the manuscript to refer to co- expression of PAX8 and WT1 when inferring the intermediate mesodermal origin of our cells.

3. I appreciate the authors' efforts in performing co-staining for GATA4, SOX9, and DHH. The colocalization of these proteins suggests that mature Sertoli cell-like cell states are induced by forced expression of NR5A1, a key driver of gonadal somatic cell differentiation.

We thank the reviewer for acknowledging these additional experiments and that in our cell system, the forced expression of NR5A1/SF1 drives the differentiation of mature Sertoli cells.

Minor comments:

Line 90: The citation of Liang et al., 2019. is not removed here.

We thank the reviewer for this correction and have removed the citation.

Fig53B, line 187: SCEA seems a typo of CSEA.

We thank the reviewer for this correction and have removed this typo in the manuscript and also in the heading of Fig53B.

line 134: The word DGE still remains here. Do the authors deliberately use the term DGE here, rather than DEG?

We thank the reviewer for this observation. In line 134, we use "DGE" to refer to the differential gene analysis method itself. However, to avoid any confusion, we have revised this to "DEG analysis (differentially expressed gene analysis)".

line 437: sPAX9 may be a typo of sPAX8.

We thank the reviewer for this comment and have corrected this typo.

Fourth decision letter

MS ID#: dev.204772R3

MS TITLE: The emergence of multiple testicular cell lineages in human stem cell-derived testis-like organoids

AUTHORS: Svenja Pachernegg, Gorjana Robevska, Lucas G.A. Ferreira, Natalie Charitakis, Jinchao Gu, Jan Terhag, Eliza Martin, Denis Bienroth, Jocelyn van den Bergen, Sean B. Wilson, Fernando J. Rossello, Ben Rollo, Melissa H. Little, Mirana Ramialison, Andrew H. Sinclair and Katie L. Ayers

Dear Dr Pachernegg,

I am happy to tell you that your manuscript has been accepted for publication in Development, pending our standard publication integrity checks.